# Cryo-EM structures of KdpFABC suggest a K$^+$ transport mechanism via two inter-subunit half-channels

C. Stock [1], L. Hielkema [2], I. Tascón [1], D. Wunnicke[1], G.T. Oostergetel [2], M. Azkargorta[3], C. Paulino [2] & I. Hänelt [1]

P-type ATPases ubiquitously pump cations across biological membranes to maintain vital ion gradients. Among those, the chimeric K$^+$ uptake system KdpFABC is unique. While ATP hydrolysis is accomplished by the P-type ATPase subunit KdpB, K$^+$ has been assumed to be transported by the channel-like subunit KdpA. A first crystal structure uncovered its overall topology, suggesting such a spatial separation of energizing and transporting units. Here, we report two cryo-EM structures of the 157 kDa, asymmetric KdpFABC complex at 3.7 Å and 4.0 Å resolution in an E1 and an E2 state, respectively. Unexpectedly, the structures suggest a translocation pathway through two half-channels along KdpA and KdpB, uniting the alternating-access mechanism of actively pumping P-type ATPases with the high affinity and selectivity of K$^+$ channels. This way, KdpFABC would function as a true chimeric complex, synergizing the best features of otherwise separately evolved transport mechanisms.

[1] Institute of Biochemistry, Biocenter, Goethe University Frankfurt, Max-von-Laue-Straße 9, 60438 Frankfurt/Main, Germany. [2] Department of Structural Biology, Groningen Biomolecular Sciences and Biotechnology Institute, University of Groningen, Nijenborgh 7, 9747 AG Groningen, The Netherlands. [3] Proteomics Platform, CIC bioGUNE, CIBERehd, ProteoRed-ISCIII, Bizkaia Science and Technology Park, Derio, Spain. These authors contributed equally: C. Stock, L. Hielkema, I. Tascón. Correspondence and requests for materials should be addressed to C.P. (email: c.paulino@rug.nl) or to I.H. (email: haenelt@biochem.uni-frankfurt.de)

Cellular K$^+$ homeostasis is fundamental for survival. In particular, prokaryotes have to cope with drastically changing environments and different external potassium concentrations. At low micromolar potassium concentrations the osmoprotective K$^+$ channels KtrAB and TrkAH, which belong to the superfamily of K$^+$ transporters (SKT), fail to maintain the internal potassium concentration. Instead, in many bacteria and archaea the primary active transport complex KdpFABC, which is highly affine for K$^+$, is produced to secure cell viability[1,2]. KdpFABC consists of four subunits[3,4] and is often referred to as P-type ATPase, since the subunit KdpB belongs to this superfamily. Usually, P-type ATPases actively pump their substrates through a single subunit composed of 8–12 transmembrane segments (TM). Transport is driven by successive ATP hydrolysis, phosphorylation, and autodephosphorylation in the three cytoplasmic domains N(ucleotide binding), P(hosphorylation), and A(ctuator)[5]. According to the Post-Albers scheme, P-type ATPases alternate between so-called E1 and E2 states. In the E1 state the substrate binds from the cytoplasm to the highly affine canonical binding site in the transmembrane domain, the N domain is loaded with Mg$^{2+}$-ATP and the conserved Asp within the P domain is phosphorylated. The subsequent E1-P to E2-P transition reorients the N, P, and A domains, locating the A domain with its TGES motif close to the phosphorylated Asp. The conformational changes are associated with rearrangements within the transmembrane segments that distort the canonical binding site and block the cytosolic access. Instead, an extracellular pathway opens, through which the substrate is released to the extracellular space. The binding of a counter-transported substrate or other effectors, stimulate the closure of the extracellular access and trigger autodephosphorylation of the Asp by the TGES motif. The protein reorientates to the E1 state, by which the counter-transported substrate is released to the cytoplasm and the transport cycle is completed[5–7]. By contrast, KdpB (7 TM) does not seem to function as common P-type ATPases. It associates with the channel-like SKT member KdpA[1,8], the periplasmatically oriented single TM subunit KdpC, and the lipid-like single spanner KdpF[9] to a unique complex that unites a P-type ATPase with a channel-like protein. Herein, KdpB hydrolyzes ATP, while K$^+$ selectivity and import has been proposed to be mediated by the channel-like subunit KdpA[10]. KdpA consists of 10 TMs, where four non-identical TM$_1$-pore helix (P)-TM$_2$ motifs (D1-D4) form the central pore comprising the selectivity filter and a pore-blocking intramembrane loop, also referred to as the gating loop (D3M$_2$)[4,11–13]. The latter is known to gate ion flux in the SKT channels TrkH and KtrB[14–17]. The currently only available crystal structure of KdpFABC, which has a Q116R mutation in KdpA leading to reduced K$^+$ affinity, was solved in a nucleotide-free E1 conformation[4]. It led to the hypothesis of a coupling mechanism between the — otherwise spatially separated — ATP hydrolysis in KdpB and K$^+$ transport through KdpA. The mechanism is based on two main structural elements: a proton-wire tunnel and a coupling helix/gating loop[4]. The authors proposed that a water-filled tunnel reaching from KdpA into KdpB with charged residues at each end could allow "communication" via a Grotthuss mechanism by moving charges along this proton-wire tunnel. This way, phosphorylation of KdpB could be initiated by the presence of K$^+$ in the selectivity filter of KdpA. Two salt bridges connect the P domain of KdpB to the distal part of helix D3M$_2$ of KdpA, also referred to as coupling helix as it, on the opposite end, forms the intramembrane loop below the selectivity filter in KdpA. Phosphorylation of Asp307 in the P domain and the accompanying large rearrangements of the subunit were, thus, suggested to pull on the coupling helix of KdpA, move the intramembrane loop and thereby open KdpA to release K$^+$ to the cytoplasm. While the intramembrane loop would serve as cytoplasmic gate, the periplasmic domain of KdpC could function as extracellular gate capping the pore of KdpA[4].

Here, we present two cryo-EM structures of wildtype KdpFABC in an E1 and an E2 state that indicate a translocation pathway for K$^+$ via two inter-subunit half-channels, integrating KdpB directly in the transport process.

## Results

**Cryo-EM structures of KdpFABC.** To elucidate the mechanism of active K$^+$ transport in KdpFABC, we aimed to determine an E2 conformation of the 157 kDa asymmetric complex using single particle cryo-electron microscopy (cryo-EM). For this purpose, KdpFABC was stabilized with its substrate K$^+$, the non-hydrolysable ATP analog AMPPCP, and the P$_i$ substitute AlF$_4^-$, similar to an approach that was used to stabilize the calcium P-type ATPase SERCA in an E2-P conformation [PDB-ID: 3B9R][18]. Notably, under this condition we were able to determine the structure of the complex in two different conformations at a resolution of 3.7 Å (state 1) and 4.0 Å (state 2), respectively (Fig. 1a, b, Supplementary Figs. 1 to 3 and Supplementary Table 1). The structures differ significantly from each other and the published crystal structure (Fig. 1, b and Supplementary Fig. 4). While all three structures align well in KdpC, KdpF and most of KdpA (Supplementary Fig. 4c), in particular the cytoplasmic domains of KdpB differ largely (Fig. 1c, d and Supplementary Fig. 5). In state 1, the A domain is rotated away from the N and P domains and the N and A domains are clearly separated. Here, the conserved TGES motif (residues 159–162 of KdpB) is spatially separated from the catalytic phosphorylation site Asp307 (Fig. 1c), which is indicative of an E1 conformation. By contrast, state 2 resembles an E2 conformation, where the A domain forms a tight interface with the N and P domains, and the TGES motif of the A domain is in place to dephosphorylate Asp307 (Fig. 1d). In agreement to these assignments, a comparison of several structures of SERCA with state 1 and state 2 resulted in the lowest RMSD values for state 1 with SERCA structures defined as E1 conformations, while state 2 agreed best with SERCA structures in an E2 conformation (Supplementary Table 2). Superpositions of the N, P, and A domains of state 1 with an E1P-ADP SERCA structure [2ZBD] and of state 2 with an E2-P structure of SERCA [3B9R], respectively, clearly show the overall similar orientation of the domains (Supplementary Fig. 6). As observed for the crystal structure, Ser162 (TGE**S**; A domain of KdpB) appears to be phosphorylated in both states (insets in Supplementary Fig. 5a and b). The high degree of phosphorylation is supported by mass spectrometric analyses of the detergent-solubilized sample before addition of conformation-specific inhibitors (Supplementary Table 3). Notably, neither in state 1 nor in state 2 Ser162-P forms a salt bridge with Lys357 and Arg363 (N domain) as observed in the crystal structure (Supplementary Fig. 5c–e). Thus, the phosphorylation of Ser162 in KdpB does not lock the A and N domains in an auto-inhibited conformation, as previously suggested[4], and it remains elusive whether the phosphorylation fully inactivates KdpFABC[4] or significantly slows down its activity.

**Inhibitor-dependent conformations probed by EPR spectroscopy.** While the assignment of the cryo-EM maps to an E1 and an E2 state is unambiguous, the local resolution of the cytoplasmic domains between 4 and 6 Å in the cryo-EM maps does not provide the required level of detail to reliably identify the binding of small ligands, impeding a further classification of the two states. In fact, we cannot rule out that each map might represent subtle different sub-states with only small structural variations in the N, P, and A domains. To evaluate whether the

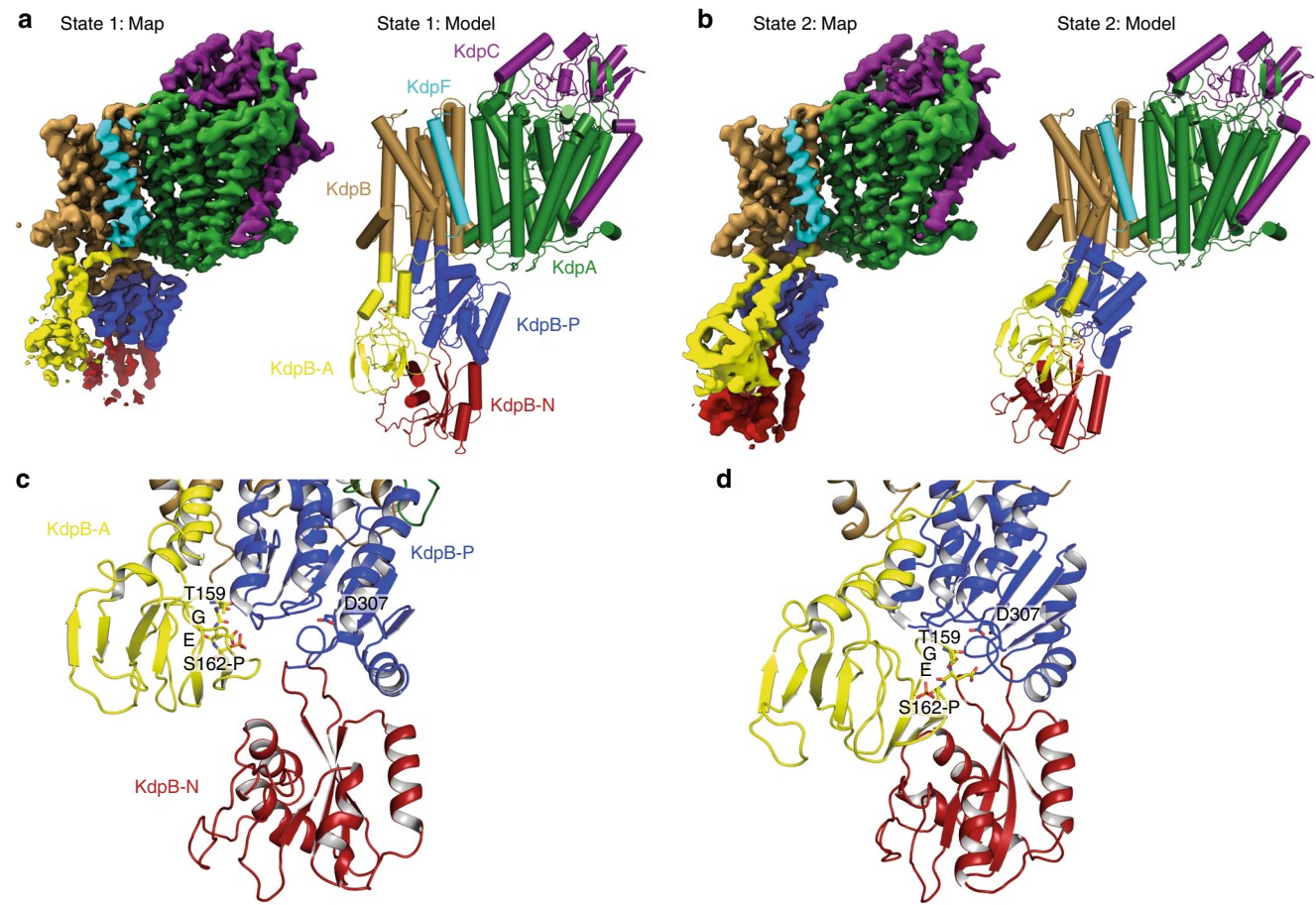

**Fig. 1** Overview of KdpFABC cryo-EM structures in different conformations. **a**, **b** Overall view of cryo-EM maps and models of the KdpFABC complex in states 1 (**a**) and 2 (**b**), models are shown in cylindrical presentation. **c**, **d** Close-up view of the N, P, and A cytoplasmic domains of KdpB in state 1 (**c**) and state 2 (**d**). Highlighted are the TGES motif, including phosphorylation at Ser162, and the catalytic phosphorylation site Asp307. Color code throughout the manuscript, unless stated otherwise, is as follows: KdpC in purple, KdpA in green, KdpF in cyan, KdpB in sand with P domain in blue, N domain in red and A domain in yellow

added ligands were essential to trap the indicated states and thus likely bound, pulsed EPR spectroscopy with a spin-labeled KdpFABC variant (KdpFAB$_{G150CR1/A407CR1}$C, labeled N and A domains) was performed (Fig. 2 and Supplementary Fig. 7). Here, the distance distribution between residues Gly150 (A domain) and Ala407 (N domain) of KdpB in the absence and the presence of defined inhibitor concentrations was determined and compared to predicted distance distributions of state 1, state 2, and the crystallographic structure[4,19]. In all measurements the rear distance distribution (marked in gray in Fig. 2 and detailed in Supplementary Fig. 7) arises from background fitting and was ignored for the interpretation of the data. In the absence of K$^+$ and inhibitors a featureless dipolar evolution trace was recorded, which resulted in a broad distance distribution between 2.2 and 4.7 nm covering state 1 and the crystallographic structure as well as several other states but not state 2 (Fig. 2, cyan traces). The addition of AMPPCP alone stabilized a conformation with a narrow distance distribution between the spin-labeled residues centered at 4 nm (Fig. 2, blue traces). Although this distance distribution does not exactly resemble the predicted distances of the state 1 cryo-EM structure, we assume that the trapped conformation represents state 1. The deviations likely arise from the poor resolution of the structure in the areas of the labeled side chains, which easily leads to the observed discrepancy in distance distributions. In addition to the main distance distribution centered at 4 nm, a

small fraction of a distance distribution below 2 nm was determined, which agrees to the calculated state 2. This indicates that KdpFABC still undergoes the complete Post-Albers cycle, in agreement with the observation that AMPPCP alone does not significantly inhibit ATPase activity (Supplementary Table 4). In contrast to AMPPCP, AlF$_4^-$ fully inhibits ATPase activity but does not stabilize a distinct conformation (Supplementary Table 4 and Fig. 2, yellow traces). Only the combination of AMPPCP and AlF$_4^-$ stabilized two main distances, one centered at 4 nm and the other at below 2 nm, which are in agreement with state 1 and state 2, respectively (Fig. 2, red traces). Based on the EPR measurements we thus postulate, that most of the particles that contributed to the reconstruction of state 1 are in an AMPPCP-bound E1 conformation, while the majority of particles that contributed to state 2 resemble an AMPPCP- and AlF$_4^-$-bound E2-P conformation. In support of this hypothesis and guided by SERCA structures [1T5S] and [3B9R], we could dock the respective molecules and coordinating magnesium ions into the structures of state 1 and state 2 (Supplementary Fig. 8a–c). AMPPCP in state 1 was placed near the highly conserved Asp307 and is likely coordinated by Mg$^{2+}$ and residues Lys308, Thr309, Thr471, Asp473, and Asn521 of the P domain. Similarly, AlF$_4^-$ in state 2 is likely coordinated by Mg$^{2+}$ and residues Asp307, Thr309, Thr471, Lys499, Asp518, and Asn521 of the P domain and Glu161 of the A domain, while AMPPCP was docked into the hypothetical

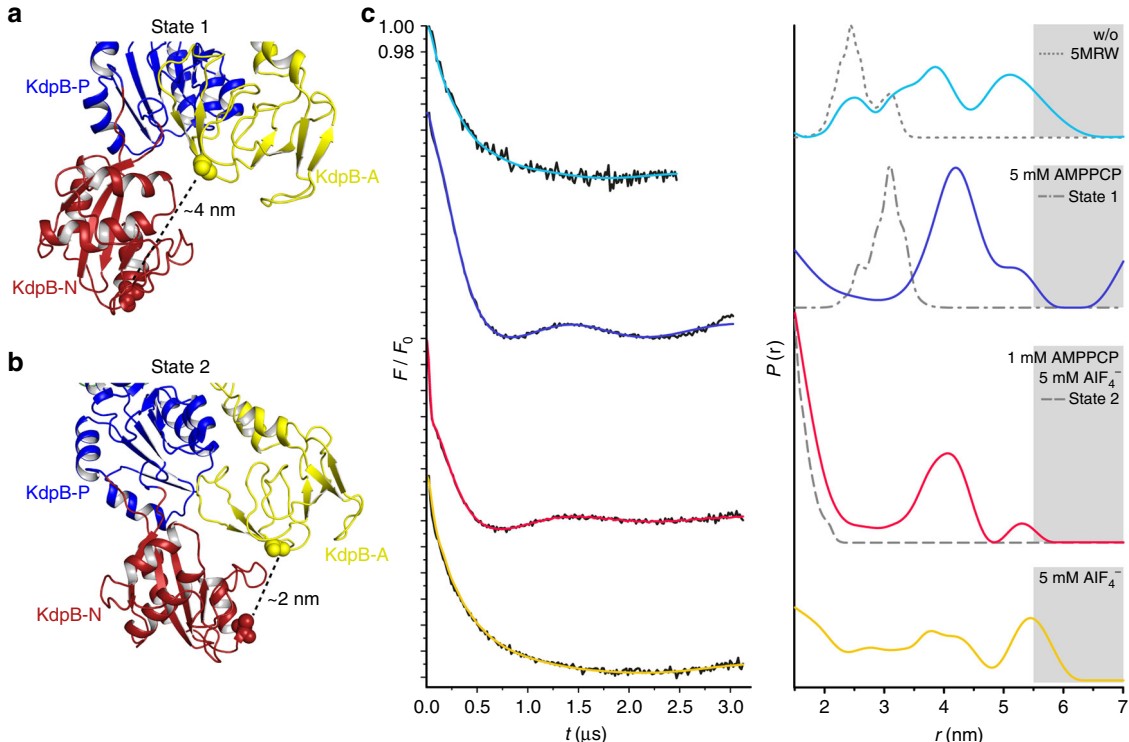

**Fig. 2** DEER measurements of KdpFABC with different inhibitors. **a**, **b** Cytoplasmic N, P, and A domains shown in state 1 (**a**) and state 2 (**b**). Respective cytoplasmic domains of variant KdpFAB$_{G150CR1/A407CR1}$C are labeled, modified residues G150 (A domain) and A407 (N domain) are shown as spheres, and approximate $C_\alpha-C_\alpha$ distances are indicated. Views are rotated 180° relative to Fig. 1c, d. **c** Left panel: dipolar evolution function F(t) with applied fit (colored lines). Right panel: area-normalized interspin distance distribution P(r) (colored lines) obtained by Tikhonov regularization. Gray areas indicate unreliable distances dependent on the dipolar evolution time. Dashed gray lines represent the predicted distance distributions of state 1, state 2 and the previously published crystal structure [5MRW][4], respectively, using the rotamer library analysis[19]

modulatory binding site formed by residues Phe377, Ser384 and Lys395 in the N domain.

**Functional states of KdpFABC**. As proposed previously, the transport cycle in KdpFABC most likely follows an inverse Post-Albers scheme, where the E1 to E2 transition is accompanied by an outward-open (state 1, E1) to inward-open transition (state 2, E2-P)[3]. Strikingly, a superposition of state 1, state 2 and the crystallographic structure [5MRW][4] of KdpFABC demonstrates the immobility of the D3M$_2$ helix and the intramembrane loop (formerly described as gating helix/loop) in KdpA, as well as of KdpC (Supplementary Fig. 9a), contradicting their previously proposed gating functions. Furthermore, in all states the potential pore of KdpA remains tightly sealed below the intramembrane loop (Supplementary Fig. 9b–e), while in other SKT members, such as TrkH[20] and KtrB[21], a large water-filled vestibule facilitates K$^+$ flux in the open state of the channels (Supplementary Fig. 9f and g). As a consequence, the calculation of a pore through KdpA towards the cytoplasm repeatedly failed for both states. Instead, all calculations revealed a pore that started from the selectivity filter, intruded horizontally into the transmembrane part of KdpA just above the intramembrane loop, and connected to the previously described tunnel[4], which links KdpA and KdpB (Fig. 3a, e and Supplementary Movie 1). Interestingly, the horizontal tunnels found in both states significantly differ in length and diameter (Supplementary Fig. 10). In state 1, a continuous tunnel starting from the selectivity filter in KdpA all the way down to the conserved canonical binding site of P-type ATPases (around Pro264) in KdpB was identified (Fig. 3a and Supplementary Fig. 10a and b). By contrast, in state 2 the inter-subunit

tunnel ends at the subunit interface of KdpA and KdpB. (Supplementary Fig. 10e and f). Instead, an additional inward-open tunnel reaching from the canonical binding site in KdpB to the cytoplasm was found (Fig. 3e). This led us to suggest an unforeseen translocation pathway for K$^+$ with outward-open and inward-open half-channels via the subunits KdpA and KdpB. In state 1, the outward-open half-channel is in principle wide enough to harbor potassium ions. Only the final segment is constrained by residues Ser579, Ile580 and Asp583 in KdpB, thus, making the canonical binding site at this point inaccessible for K$^+$ (Supplementary Fig. 10a and b). However, the tunnel identified in the crystal structure showed a diameter broad enough for K$^+$ passage[4], supporting the idea of partially hydrated K$^+$ moving from the selectivity filter in KdpA all the way to the canonical binding site in KdpB (Supplementary Fig. 10c and d). In contrast, the inward-facing half-channel is completely closed in state 1. Residues Ser272 within TM4 and Glu296 in the P domain restrict tunnel formation and the cytoplasmic exit is tightly sealed by inter-subunit salt bridges between Arg400 and Gln513 in KdpA and Asp300, Asp302 and Gly510 in the P domain of KdpB, respectively (Fig. 4a). In fact, these salt bridges are those previously proposed to mediate the opening of the intramembrane loop in KdpA[4]. In state 2, the outward-facing half-channel is tightly sealed at the interface of KdpA and KdpB around residues Phe386, Leu389, Ile421, Leu422, Val538, and Leu541 of KdpA and Leu228 and Val231 of KdpB (Fig. 3e, f and Supplementary Fig. 10e and f), restricting the entrance of K$^+$ to the canonical binding site. Furthermore, Asp583 and Pro264 close the binding site towards the outward-facing tunnel (Fig. 4c). Instead, rearrangements of the P domain disrupt the above-mentioned inter-subunit salt bridges at the cytoplasmic side, well separating

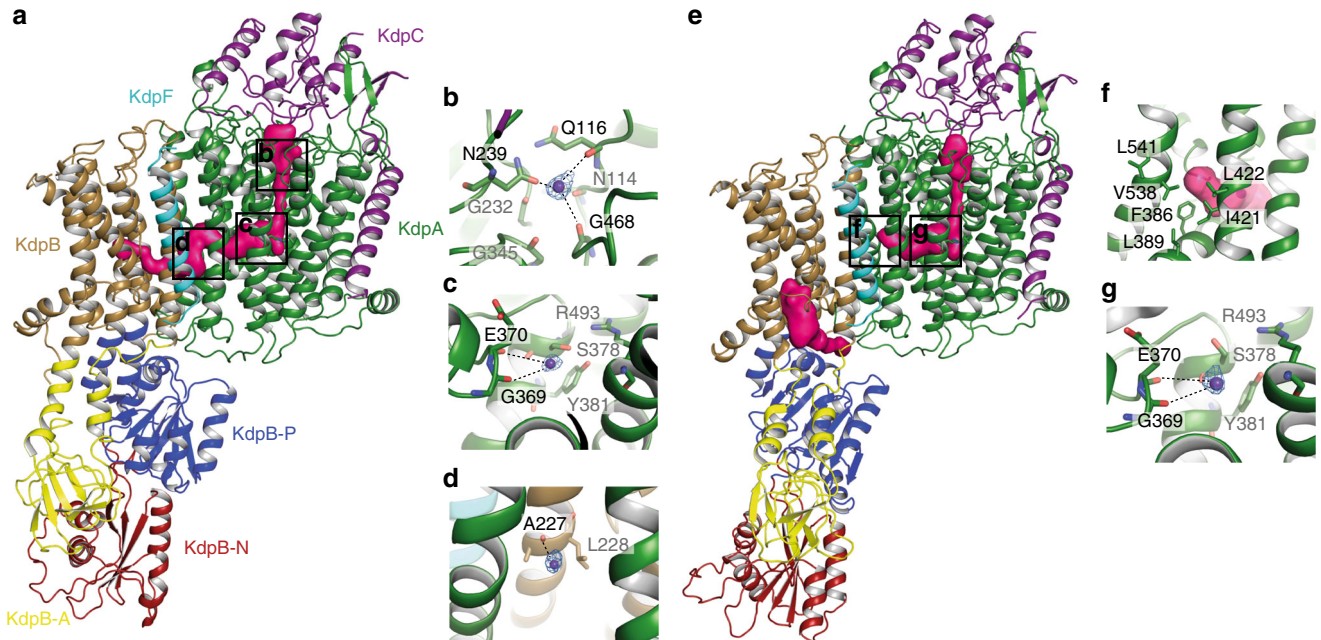

**Fig. 3** Outward-open and inward-open half-channels in states 1 and 2 of KdpFABC. **a** Entrance tunnel in state 1 covering the selectivity filter and the inter-subunit tunnel between KdpA and KdpB. **b–d** Magnification of the K$^+$ binding sites inside the entrance tunnel in state 1 shown with corresponding cryo-EM density map for the potassium ions sharpened with b-factors of −205 Å$^2$ at 5.5 σ (**b**), 6.0 σ (**c**), and 6.5 σ (**d**). Coordinating residues are represented as sticks. **e** Blocked entrance tunnel at the KdpA-KdpB interface and open exit pathway between KdpA and the P domain of KdpB in state 2. **f** Close-up view of the entrance tunnel blockage in state 2. Tunnel-blocking residues of KdpA are represented as sticks. **g** Magnification of the K$^+$ binding site inside the residual entrance tunnel in state 2 shown with cryo-EM density map for K$^+$ sharpened with a b-factor of -195 Å$^2$ at 5.5 σ. Coordinating residues represented as sticks. Potassium ions are shown as dark purple spheres. Entrance and exit tunnel surface representations (pink densities) were calculated with Hollow[61]

subunits KdpA and KdpB (Fig. 4b). TM2 and TM4 of KdpB moved by 5 and 15 degrees and the side chains of Asn624 and Thr265 reoriented, opening the tunnel from the canonical binding site to the cytoplasm (Fig. 4c-e). We suggest that a reshaping of the ion binding site in state 2 lowers the affinity for K$^+$, which triggers its release into the cytoplasm. In particular, the observed movement of Lys586 during the E1/E2 transition might suffice to push K$^+$ off the binding pocket (Fig. 4c). Thus, the motion of residues Asp583 and Lys586 in TM5 of KdpB, which were previously described as regulatory dipoles[22,23], might be crucial for the transport mechanism in KdpFABC and account for the protein-bound charge movements measured in electrophysiological experiments[24,25]. The central role of Asp583 and Lys586 proposed here might also account for the striking phenotype observed for D583A and D583K/K586D mutants, which both abolished transport but showed K$^+$-uncoupled ATPase activity[22,23,26]. We speculate that the removal of the negative charge at position 583 mimics bound K$^+$ and, consequently, might be sufficient to stimulate ATPase activity. Notably, the location of the inward-open tunnel differs from the common exit site found in other P-type ATPases, which might be a consequence of the interaction with KdpA or due to the minimal structure of KdpB, which lacks three transmembrane helices when compared to SERCA.

In support of the proposed translocation pathway three defined densities are observed inside the outward-open half-channel of state 1, and one in the residual half-channel of state 2 (Fig. 3a–e, g and Supplementary Fig. 11a to d). Given the fact that KdpFABC is highly selective for K$^{+27}$ and that the data was recorded in the presence of 1 mM KCl, we have assigned these densities to potassium ions. Consequently, in state 1 one K$^+$ would be located at the outer S1 position of the selectivity filter (Fig. 3b and Supplementary Fig. 11a) and two within the inter-subunit tunnel (Fig. 3c, d and Supplementary Fig. 11b and c). K$^+$ in the S1

position is partially coordinated by the side chain of Asn239 and the hydroxyl groups of residues Gln116 and Gly468 of KdpA (Fig. 3b and Supplementary Fig. 11a). The potassium ions within the inter-subunit tunnel are located in rather spacious sections (radii up to 2.5 Å), suggesting a partial coordination by water molecules. Furthermore, the hydroxyl groups of Asp370 and Gly369 provide direct coordination for the second ion, while only the hydroxyl group of Ala227 seems to be in direct contact with the third ion, which shows the weakest density. In state 2, the only density potentially representing K$^+$ was found within the remaining tunnel in KdpA (Fig. 3g and Supplementary Fig. 11d). In addition, we found an unassigned density at the interface of KdpA and KdpB that most likely corresponds to a bound lipid and might play a role in tunnel formation and ion propagation (Supplementary Fig. 11e and f). Though, molecular dynamics (MD) simulations, anomalous signals from X-ray crystallography or other formal proofs are required to elucidate the exact K$^+$ binding sites and ion propagation mechanism.

## Discussion

The presented E1 and E2 structures led us to propose a mechanism for active transport of K$^+$ via KdpFABC (Fig. 5 and Supplementary Movie 1). Based on sequence alignments with KtrB and TrkH and functionally impaired KdpFABC variants with mutations in the selectivity filter, KdpA has been assumed to be the K$^+$-translocating subunit. However, no transport assays supporting this assumption are, to our knowledge, available. Further, in contrast to KtrB and TrkH we found that the expression of KdpA subunit alone does not support the uptake of potassium ions, as one would expect for a protein with preserved channel-like features (Supplementary Fig. 12). Instead, we propose that not solely the channel-like KdpA subunit facilitates K$^+$ translocation, but rather the combination of two joined half-channels formed by KdpA and KdpB. Here, substrate occlusion

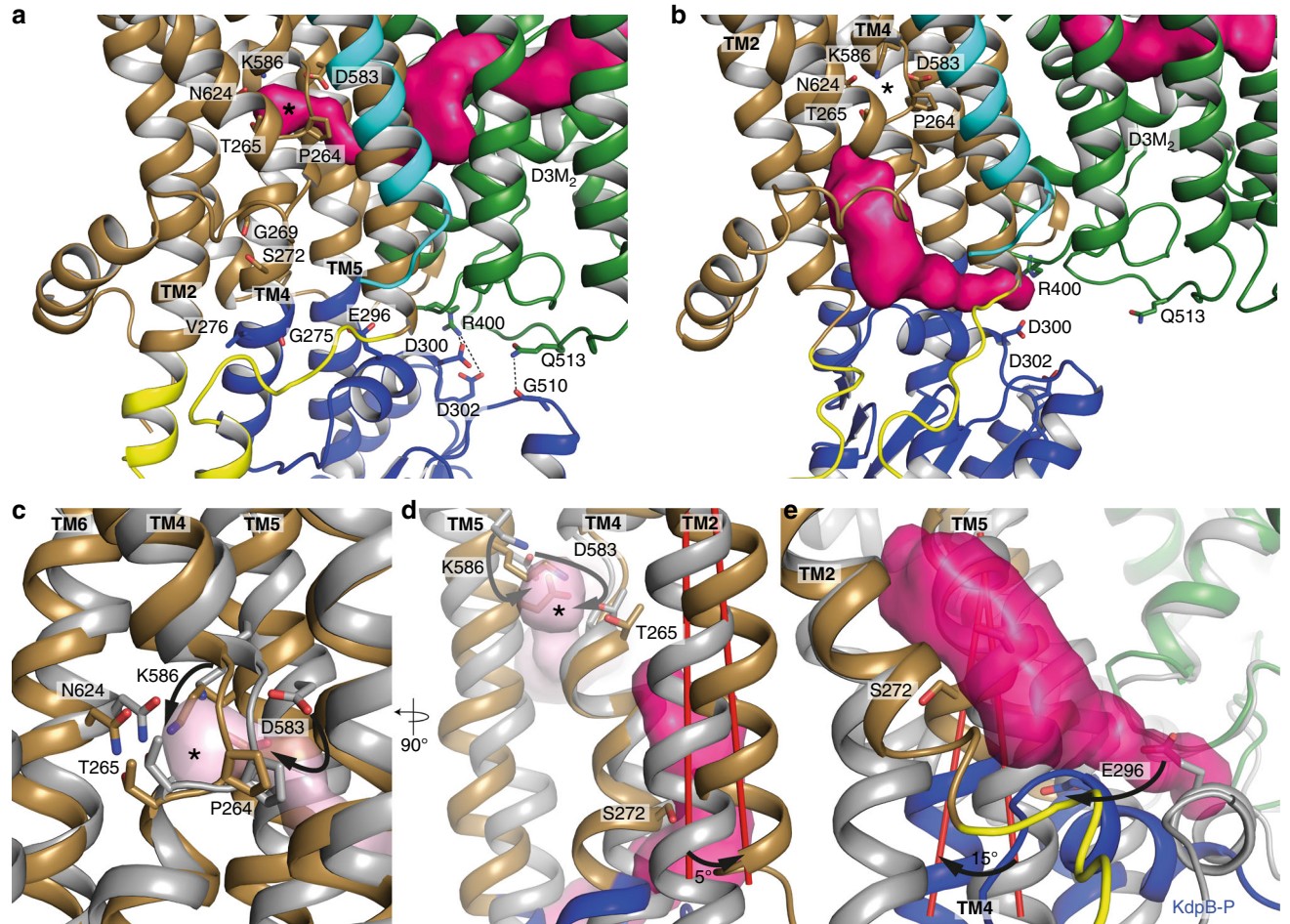

**Fig. 4** Conformational changes during state 1 to state 2 transition. **a** Blocked cytoplasmic exit site in state 1. Residues Arg400 (KdpA) and Asp300, Asp302 (KdpB) and Gln513 (KdpA), and Gly510 (KdpB), respectively, form salt bridges at the P domain-KdpA interface. Further blockage of the exit tunnel is achieved by residues Gly269, Ser272, Gly275, Val276 and Glu296 of KdpB. **b** Open exit tunnel in state 2 ranging from the canonical binding site (Pro264, Thr265, Asp583, Lys586, and Asn624) to the cytoplasmic exit site, where residues Arg400 (KdpA) and Asp300, Asp302 (KdpB) as well as Gln513 (KdpA) and Gly510 (KdpB) do not interact. **c** Close-up superposition of the canonical binding site in state 1 (gray) and state 2 (color) with entrance tunnel (light pink density). Key residues Pro264, Asn624, Thr265, Asp583, and Lys586 undergo large conformational changes distorting the binding site. **d**, **e** Superposition of transmembrane helices of KdpB in state 1 (in gray) and state 2 (color) highlighting the opening of the exit tunnel. TM2 rotates by 5° (**d**) and TM4 by 15° (**e**) opening up the tunnel towards the canonical binding site. In particular the delocalization of residues Ser272 within TM4 and Glu296 in the P domain of KdpB allow the tunnel formation. Tunnel densities depicted in light and dark pink (entrance and exit tunnel, respectively). Asterisk (*) in **a**–**d** indicates the canonical binding site. Movement of the conserved residues Asp583 and Lys586 is highlighted with black and gray arrows in **c** and **d**, respectively. View in **d** is rotated by 90° to the left from figures **a**–**c**

can occur at the canonical binding site of the P-type ATPases KdpB, while the system acquired a high K$^+$ selectivity and affinity by 'hijacking' a channel's selectivity filter, as found in KdpA. Although, the selectivity filter itself warrants selective ion binding, the pathway through both subunits might account for the higher selectivity found in KdpFABC when compared to other SKT members. Interestingly, while mutations in for example KtrAB and high-affinity potassium transporters (HKT) shifted the selectivity towards Na$^{+}$[28–31], only mutations that led to an additional transport of Rb$^+$ and NH$_4$$^+$—which have a similar ion radius as K$^+$—could be identified for KdpFABC[11–13,32]. Likewise, NH$_4$$^+$ and Rb$^+$ are known substituents for K$^+$ in Na$^+$/K$^+$ ATPase[33]. On the other hand, to which extend the selectivity filter, the tunnel and the canonical binding contribute to the observed high affinity of KdpFABC will require additional studies. Finally, we hypothesize that K$^+$ translocation occurs via a tightly controlled knock-on like mechanism[34,35], by which an unknown number of ions propagates through the outward-open half-channel to finally bind at the canonical binding site in KdpB.

The release of K$^+$ from the KdpB subunit into the cytoplasm is the result of reduced binding affinity due to the distortion of the canonical binding site by conformational changes. Arg493, located within the entrance tunnel in KdpA, Asp583, and Lys586 at the canonical binding site in KdpB and Asp300, located at the cytoplasmic end of the exit tunnel[22,23,36], were shown to be crucial for activity, supporting their putative role in the translocation network. In fact, the proposed mechanism in which the ion channel pore remains closed and potassium ions are redirected through the P-type ATPase subunit makes it easier to envision how potassium ions are actively pumped by the complex against a concentration gradient as high as 10$^4$[37]. Notably, the alternating access of the binding site with outward-facing E1 and inward-facing E2 states suggested here is reversed in comparison to classical P-type ATPases[38]. Although this hypothesis is supported by several other studies[3,4,39], there is also evidence in favor of the classical reaction cycle[24,40,41]. Particularly, our model contradicts the functional studies from Siebers and Altendorf[40], in which the KdpFABC complex was maximally phosphorylated upon

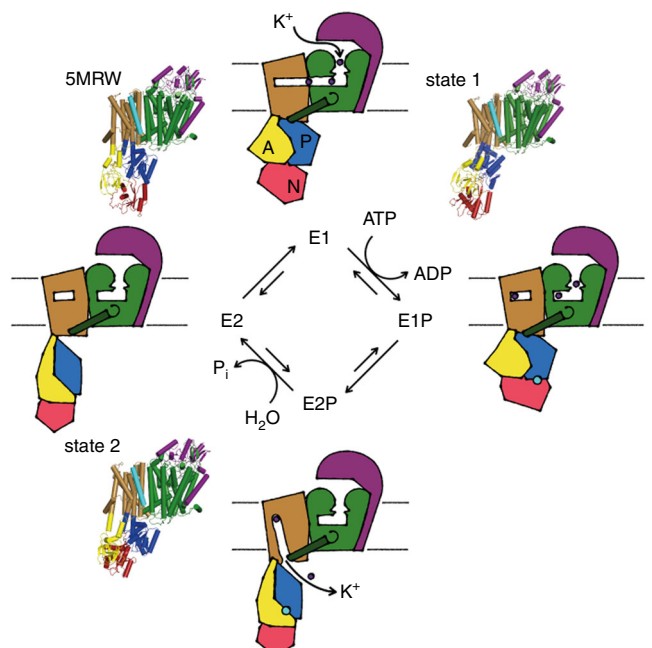

**Fig. 5** Proposed transport cycle of KdpFABC according to a Post-Albers scheme. In the E1 state, the selectivity filter in KdpA selectively binds K⁺, from where it enters the outward-open half-channel by a tightly controlled knock-on mechanism. Once K⁺ has reached its canonical binding site in KdpB, the tunnel closes at the interface of KdpA and KdpB occluding further passage of K⁺. Simultaneously, ATP is hydrolyzed to phosphorylate Asp307 in KdpB's P domain (E1P). Rearrangements within the cytoplasmic domains bring the TGES motif of the A domain in close proximity to Asp307 in the P domain (E2-P). Here, the canonical binding site is distorted, the salt bridges between KdpA's coupling helix and KdpB's P domain are disrupted and the inward-open half-channel is formed triggering the release of K⁺ into the cytoplasm. Dephosphorylation of Asp307 and $P_i$ release reset the transporter to the E1 state via an E2 apo-state. Structures of state 1 and state 2 as well as the crystal structure [5MRW][4] are positioned at their presumed location in the reaction cycle. KdpA green with coupling helix dark green; KdpB sand with N domain red, P domain blue, A domain yellow and phosphorylated Asp307 cyan; KdpC purple; K⁺ dark purple; KdpF is removed for simplicity

addition of ATP in the absence of K⁺, while the addition of K⁺ induced dephosphorylation. Thus, further functional and structural data are required to clarify the exact transport cycle. Another uncertainty is the functional role of KdpC. In light of absent conformational changes and KdpC's proximity to the selectivity filter, we suggest that it might function similar to β subunits of Na⁺/K⁺ ATPase, and gastric H⁺ ATPase (Supplementary Fig. 13) and increase K⁺ affinity[42], as speculated 30 years ago[43].

In summary, we propose that even at very low concentrations, potassium ions are attracted with high affinity to the selectivity filter in KdpA and move along the outward-open half-channel in the E1 state. Tightly bound K⁺ at the canonical binding site in KdpB triggers the first state transition; ATP is hydrolyzed and Asp307 in the P domain is phosphorylated. The occlusion of K⁺ within KdpB in the E1P state is followed by prominent reorientations of particularly the A domain to position the TGES motif for dephosphorylation of Asp307. The P domain moves away from the $D3M_2$ (coupling) helix of KdpA, thereby breaking the salt bridges, reorienting TMs 2 and 4 in KdpB and disrupting the canonical binding site. The resulting E2-P state opens an inward-open half-channel at the interface of KdpB and KdpA, releasing K⁺ from the binding site to the cytoplasm. In two final

steps Asp307 is dephosphorylated (E2) and the cytoplasmic domains as well as the half-channels reorient to regenerate KdpFABC for a new transport cycle (Fig. 5). The here-solved structures neither represent fully outward-open nor fully inward-open states but most likely trapped intermediate conformations. We suggest that state 1 is a partially K⁺-loaded E1 state, which in a transport cycle follows the E1 crystal structure with a single K⁺ in the S3 position. State 2 likely represents an E2-P state after ion release, in which the inward-open half-channel is already partially collapsed.

The here proposed transport mechanism for the unique KdpFABC complex combines key features of a primary active P-type ATPase with the high affinity and selectivity of an ion channel, providing insights on how the complex is able to efficiently pump potassium ions despite low external concentrations. Our data picture a true chimeric complex between a transporter and a channel. KdpFABC demonstrates how in the course of evolution conserved protein architectures not only evolved from one another but can merge together to adapt to different environmental and cellular requirements.

## Methods

**Cloning and protein production and purification.** *kdpFABC* and its cysteine-free mutant (provided by J.C. Greie, Osnabrück, Germany) from *Escherichia coli* were cloned into FX-cloning vector pBXC3H resulting in pBXC3H-KdpFABC and pBXC3H-KdpFABCΔCys, respectively[44]. pBXC3H-KdpFABCΔCys-KdpB:G150C/A407C was created by site-directed mutagenesis based on the latter. Wildtype *kdpFABC* encoded by plasmid pBXC3H-KdpFABC was expressed in *E. coli* strain C43[45] (Lucigen) aerobically in full media (10% trypton, 10% NaCl, 5% yeast extract) at 37 °C by the induction with 0.002% L-arabinose in the late-exponential growth phase. One hour after induction the cells were harvested by centrifugation at 5000xg and 4 °C. Cells were resuspended in 50 mM Tris-HCl pH 7.5, 10 mM MgCl₂, 1 mM DTT, 10% glycerol, 2 mM EDTA and 0.5 mM PMSF prior to cell disruption (Stansted, pressure cell homogenizer). After cell fractionation, membrane solubilization (1% n-Dodecyl-β-D-Maltopyranoside (DDM), in 50 mM Tris-HCl pH 7.5, 10 mM MgCl₂, 10% glycerol, 0.5 mM PMSF) was carried out at 4 °C overnight at a protein concentration of 10 mg ml⁻¹. The solubilized protein fraction was supplemented with 10 mM imidazole, 150 mM NaCl and 0.5 mM PMSF, immobilized on a Ni Sepharose™ 6 fast Flow column (GE Healthcare) for one hour, and unbound proteins were removed by washing with 50 column volumes wash buffer (50 mM Tris-HCl pH 7.5, 20 mM MgCl₂, 150 mM NaCl, 10% glycerol, 0.025% DDM) containing 30 mM imidazole. For specific KdpFABC elution, on-column cleavage with 1 mg ml⁻¹ 3C protease in 1–2 column volumes wash buffer supplemented with 0.1 mM PMSF was performed at 4 °C for one hour. KdpFABC was transferred into AIEX buffer (10 mM Tris-HCl pH 8, 10 mM MgCl₂, 10 mM NaCl and 0.025% DDM (≥99%, highly purified, GLYCON Biochemicals GmbH) with Zeba™ Spin desalting columns (7 K MWCO, Thermo Scientific) and loaded on a HiTrap Q HP column (GE Healthcare) attached to an ÄKTA pure system (GE Healthcare). Gradual elution (10–500 mM NaCl in AIEX buffer) was applied, a single peak collected and subjected to size exclusion chromatography on a preparative Superdex 200 10/300 GL column (GE healthcare), pre-equilibrated with SEC buffer (10 mM Tris-HCl pH 8, 10 mM MgCl₂, 10 mM NaCl and 0.012% DDM (≥99%, highly purified, GLYCON Biochemicals GmbH). Variant KdpFAB$_{G150C/A407C}$C was produced aerobically at 37 °C in *E. coli* strain LB2003[46] (available from Hänelt upon request) in minimal medium[21] containing 3 mM KCl upon induction with 0.002% L-arabinose directly after inoculation. The protein purification until the binding of the protein to the IMAC resin was performed as described for the wildtype. Subsequently, by washing with 50 column volumes wash buffer containing 30 mM imidazole and 5 mM β-mercaptoethanol unbound proteins were removed and cysteines reduced. Afterwards, both reducing agent and imidazole were removed by washing with 15 column volumes wash buffer for adjacent spin-labeling. Spin-labeling with 1 mM MTSSL (1-oxyl-2,2,5,5- tetramethylpyrrolidin-3-yl)methylmethanethiosulfonate spin label, Toronto Research Chemicals) in wash buffer was performed on-column overnight at 4 °C and excessive unbound spin label was removed with 30 column volumes of wash buffer. Spin-labeled KdpFABC was eluted from the column with three times one column volume wash buffer supplemented with 250 mM imidazole and 0.1 mM PMSF. Finally, protein-containing fractions were subjected to size exclusion chromatography under wildtype conditions with 0.025% DDM. Purified, spin-labeled KdpFAB$_{G150CR1/A407CR1}$C was concentrated to 4−7 mg ml⁻¹ and 14% deuterated glycerol (v/v), inhibitors as indicated and 1 mM KCl were added for further pulsed EPR measurements.

**Cryo-EM sample preparation and data acquisition.** Freshly purified KdpFABC complex at a concentration of 3.1 mg ml⁻¹ in the presence of inhibitor mix (1 mM

AMPPCP, 5 mM AlF$_4^-$, 1 mM KCl) were applied with a volume of 2.8 µl on holey-carbon cryo-EM grids (Quantifoil Au R1.2/1.3, 200 and 300 mesh), which were prior glow-discharged at 5 mA for 20 s. Grids were blotted for 3–5 s in a Vitrobot (Mark 3, Thermo Fisher) at 20 °C temperature and 100% humidity, subsequently plunge-frozen in liquid ethane and stored in liquid nitrogen until further use. Cryo-EM data were collected on a 200 keV Talos Arctica microscope (Thermo Fisher) using a post-column energy filter (Gatan) in zero-loss mode, using a 20 eV slit, a 100 µm objective aperture, in an automated fashion using EPU software (Thermo Fisher) on a K2 summit detector (Gatan) in counting mode. Cryo-EM images were acquired at a pixel size of 1.012 Å (calibrated magnification of 49,407×), a defocus range from −0.3 to −3 µm, an exposure time of 9 sec and a sub-frame exposure time of 150 ms (60 frames), and a total electron exposure on the specimen level of about 52 electrons per Å$^2$. Best regions on the grid were screened with a self-written script to calculate the ice thickness and data quality was monitored on-the-fly using the software FOCUS[47].

**Cryo-EM image processing**. A total of 7327 dose-fractionated cryo-EM images were recorded and subjected to motion-correction and dose-weighting of frames by MotionCor2[48]. The CTF parameters were estimated on the movie frames by ctffind4.1[49]. Bad images showing contamination, a defocus below or above −0.3 and −3.0 µm or a bad CTF estimation were discarded, resulting in 5828 images used for further analysis with the software package RELION2.1[50]. About 1100 particles were picked manually to generate 2D references, which were improved in several rounds of autopick. The final round of autopick yielded an initial set of 826,403 particles. False positives or particles belonging to low-abundance classes were removed in several rounds of 2D classification, resulting in 535,981 particles. Due to the large conformational differences between both states, the full dataset was further cleaned by two independent 3D classifications against references obtained for state 1 and state 2. Particles belonging to the best classes of both runs were merged and duplicates subtracted, resulting in 466,198 particles that were subjected to a multi-reference 3D classification with no image alignment. The dataset was from here on treated separately, with about 60% (283,307 particles) in state 1 and about 40% (182,890 particles) in state 2. Both datasets were subjected to another round of 3D classification, which resulted in a cleaned-up dataset of 219.897 particles for state 1 and 104.786 particles for state 2. The final map for state 1 had a resolution of 4.3 Å before masking and 3.7 Å after masking and was sharpened using an isotropic b-factor of −154 Å$^2$. For manual inspection and some of the figures a b-factor of −205 Å$^2$ was used. The final map for state 2 had a resolution of 4.7 Å before masking and 4.0 Å after masking and was sharpened using an isotropic b-factor of −147 Å$^2$. For manual inspection and some of the figures a b-factor of −195 Å$^2$ was used. Particles were initially extracted with a box size of 320 and later with a box size of 240 pixels. Initial classification steps were performed with 3.2-fold binned data. For 3D classification and refinement, a map generated from the crystal structure [5MRW][4] was used as reference for the first round, and the best output class was used in subsequent jobs in an iterative way. No symmetry was imposed during 3D classification or refinement. The approach of focused refinement, where the less-resolved detergent micelle was subtracted from the particle images, did not improve resolution[51]. Local resolution estimates were calculated by RELION. All resolutions were estimated using the 0.143 cut-off criterion[52] with gold-standard Fourier shell correlation (FSC) between two independently refined half maps. During post-processing, the approach of high-resolution noise substitution was used to correct for convolution effects of real-space masking on the FSC curve[53].

**Model building and validation**. The crystal structure of KdpFABC [5MRW][4] was split into individual subunits. Additionally, KdpB was divided into four parts: KdpB-TM (residues 9–88, 216–274, and 570–682), KdpB-P (residues 275–314 and 451–569), KdpB-N (residues 315–450) and KdpB-A (residues 89–215). Initially, all fragments were docked into the two obtained cryo-EM maps using UCSF Chimera[54]. The connections between the four KdpB segments were modeled manually in Coot[55]. The initial model, for each data set, was then subjected to an iterative process of real space refinement using Phenix.real_space_refinement with secondary structure restraints[56,57] followed by manual inspection and adjustments in Coot[55]. Potassium ions in the outward-open half-channel were modeled into the cryo-EM maps. The final models were refined in real space with Phenix.real_space_refinement with secondary structure restraints[56,57]. For validation of the refinement, FSCs (FSC$_{sum}$) between the refined models and the final maps were determined. To monitor the effects of potential over-fitting, random shifts (up to 0.5 Å) were introduced into the coordinates of the final model, followed by refinement against the first unfiltered half-map. The FSC between this shaken-refined model and the first half-map used during validation refinement is termed FSC$_{work}$, and the FSC against the second half-map, which was not used at any point during refinement, is termed FSC$_{free}$. The marginal gap between the curves describing FSC$_{work}$ and FSC$_{free}$ indicate no over-fitting of the model. The geometries of the atomic models were evaluated by MolProbity[58]. Tunnels and pore radii were calculated using Caver_Analyst[59] and HOLE[60] softwares, respectively. Surface representations of the tunnels and pores were obtained with Hollow[61]. Comparisons of KdpB with different structures of SERCA were done in COOT[55]. For AMPPCP and AlF$_4^-$ docking the crystallographic structures of SERCA in an E1 state [1T5S] and E2-P state [3B9R] were used as guides. All figures were prepared using UCSF Chimera[54] and Pymol[62].

**ATPase assay**. The ATPase activity of purified KdpFABC complexes (µmol P$_i$ mg$^{-1}$ min$^{-1}$) was determined by the malachite green assay[63] at 37 °C. In brief, 0.25 µg of protein were pre-incubated at the indicated conditions for 5 min at 4 °C. Reactions were started by the addition of 2 mM ATP and carried out at 37 °C for 5 min.

**EPR sample preparation and data acquisition and analysis**. Pulsed EPR measurements were performed at Q band (34 GHz) and −223 °C on an Elexsys 580 spectrometer (Bruker). Therefore, 15 µl of the freshly prepared samples were loaded into EPR quartz tubes with a 1.6 mm outer diameter and shock frozen in liquid nitrogen. During the measurements, the temperature was controlled by the combination of a continuous-flow helium cryostat (Oxford Instruments) and a temperature controller (Oxford Instruments). The four-pulse DEER sequence was applied[64] with observer pulses of 32 ns and a pump pulse of 13–18 ns. The frequency separation was set to 70 MHz and the frequency of the pump pulse to the maximum of the nitroxide EPR spectrum. Validation of the distance distributions was performed by means of the validation tool included in DeerAnalysis[65] and varying the parameters "Background start" and "Background density" in the suggested range by applying fine grid. A prune level of 1.15 was used to exclude poor fits. Furthermore, interspin distance predictions were carried out by using the rotamer library approach included in the MMM software package[19]. The calculation of the interspin distance distributions is based on the cryo-EM structures of state 1, state 2 and the crystal structure [5MRW][4] for the comparison with the experimentally determined interspin distance distributions.

**Tryptic digestion**. Gel bands of purified KdpB were washed in Milli-Q water. Reduction and alkylation were performed using dithiothreitol (10 mM DTT in 50 mM ammonium bicarbonate) at 56 °C for 20 min, followed by iodoacetamide (50 mM iodoacetamide in 50 mM ammonium bicarbonate) for further 20 min in the dark. Gel pieces were dried and incubated with trypsin (12.5 µg ml$^{-1}$ in 50 mM ammonium bicarbonate) for 20 min on ice. After rehydration, the trypsin supernatant was discarded. Gel pieces were hydrated with 50 mM ammonium bicarbonate, and incubated overnight at 37 °C. After digestion, acidic peptides were cleaned with TFA 0.1% and dried out in a RVC2 25 speedvac concentrator (Christ). Peptides were resuspended in 10 µl 0.1% FA and sonicated for 5 min prior to analysis.

**NanoLC-MS/MS and data analysis**. Peptide mixtures obtained from trypsin digestion were separated by online nanoLC and analyzed by electrospray tandem mass spectrometry. Peptide separation was performed on a nanoAcquity UPLC system (Waters). Samples were loaded onto a Symmetry 300 C18 UPLC Trap column, 180 µm x 20 mm, 5 µm (Waters). The pre-column was connected to a BEH130 C18 column, 75 µm x 200 mm, 1.7 µm (Waters) equilibrated in 3% acetonitrile and 0.1% FA, and peptides were eluted at 300 nl min$^{-1}$ using a 30 min linear gradient of 3–50% acetonitrile directly onto the nanoelectrospray ion source of a Synapt G2Si ESI Q-Mobility-TOF spectrometer (Waters) equipped with an ion mobility chamber (T-Wave-IMS). All analyses were performed in positive mode ESI. Data were post-acquisition lock mass corrected using the double charged monoisotopic ion of [Glu1]-Fibrinopeptide B. Accurate mass LC-MS data were collected in HDDA mode that enhances signal intensities using the ion mobility separation step. Searches were performed using Mascot Search engine (Matrix Science) on Proteome Discoverer 1.2. software (Thermo Electron). Carbamido-methylation of Cys was considered as fixed modification, and oxidation of Met and phosphorylation of Ser, Thr, Tyr, and Asp as variable modification. PhosphoRS[66] was used in the workflow in order to calculate the probabilities for all possible phosphorylation sites. 10 ppm of peptide mass tolerance, and 0.2 Da fragment mass tolerance were adopted as search parameters. Spectra were searched against an E. coli Uniprot/Swissprot database (2016_09). Only peptides passing the $p < 0.01$ high-confidence cutoff were considered as reliable hits for further discussion.

**Growth complementation assay**. The growth complementation assays were performed as previously described[29]. In brief, His-tagged versions of KdpA, KdpFABC and KtrB, respectively, were expressed in E. coli LB2003, a strain lacking all endogenous K$^+$ uptake systems. LB2003 transformed with empty vector pBAD18 served as negative control. Growth curves were recorded for 24 h at different K$^+$ concentrations (1–115 mM referred to as K1–K115). At K10 and below the strain only grows sufficiently, if the expressed protein complements the lacking transport systems. Protein production was confirmed by Western blotting analysis of a K30 sample after 24 h using an anti-His antibody from mouse (dilution 1:3000, Sigma-Aldrich, cat.no. H1029) and secondary anti-mouse IgG-peroxidase antibody produced in goat (dilution 1:20,000, Sigma-Aldrich, cat.no. A2554).

**Reporting summary**. Further information on research design is available in the Nature Research Reporting Summary linked to this article.

## Data availability

Data supporting the findings of this manuscript are available from the corresponding authors upon reasonable request. A reporting summary for this Article is available as a Supplementary Information file. The three-dimensional cryo-EM densities of KdpFABC have been deposited in the Electron Microscopy Data Bank under accession numbers EMD-0257 for state 1 and EMD-0258 state 2, respectively. The depositions include maps calculated with higher $b$-factors, both half-maps and the mask used for the final FSC calculation. Modeled coordinates have been deposited in the Protein Data Bank under accession numbers 6HRA for state 1 and 6HRB for state 2, respectively.

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

## Acknowledgements

I.H. thanks Prof. Karlheinz Altendorf (supported by the Friedel & Gisela Bohnenkamp-Stiftung) and Dr. Jörg-Christian Greie for providing her the materials and insights for the work on KdpFABC. Prof. Thomas Prisner and Dr. Burkhard Endeward are acknowledged for their support on the EPR measurements. C.S. thanks Jakob Merlin Silberberg for assisting with cell growth. C.P. thanks Michiel Punter for the support in setting up the image processing cluster. This work was supported by the German Research Foundation via Emmy Noether grant HA 6322/3-1 and the Cluster of Excellence Frankfurt (Macromolecular Complexes) to I.H.; and by the NWO Veni grant 722.017.001 and Marie Skłodowska-Curie Individual Fellowship 749732 to C.P.

## Author contributions

C.S. cloned, expressed and purified KdpFABC and performed ATPase and complementation assays; L.H., G.T.O. and C.P. prepared the sample for cryo-EM and collected electron microscopy data. L.H. and C.P calculated and validated the cryo-EM maps; I.T. modeled the structures and analyzed the structural features; D.W. performed EPR measurements; M.A. performed mass spectrometry; C.S., I.T., C.P. and I.H. interpreted the structures and wrote the manuscript; C.P. and I.H. conceived the project.

## Additional information

**Competing interests:** The authors declare no competing interests.

