## [Peer Review File · Nature Communications]

Reviewers' Comments:

Reviewer #1:

Remarks to the Author:

In this interesting and important contribution cryo-electron microscopy was used to elucidate the structure of the KdpFABC complex. Two new structures with a resolution of 3.7 Å and 4.0 Å could be presented. In contrast to the previous single structure of the complex with almost atomic resolution that was gained by X-ray diffraction analysis from protein crystals, the cryo-EM structures are obtained from single molecules, solubilized in DDM, which did not need to make possibly distorting crystal contacts.

The analyzed samples contained KdpFABC complexes in the presence of saturating concentrations of KCl, MgCl₂, AMPPNP, and AlF₄⁻. These substrates, partly indicated as 'inhibitors', were able to bind to the complex in different combinations and induced diverse states. During the analysis procedure the authors confined themselves to focusing on two states, named 'state 1' and 'state 2'.

In contrast to assumptions of the ion transport mechanism presented in previous papers, now solid evidence is presented that the pathway of the ions is not restricted to the K⁺-channel like KdpA subunit but leads through both the KdpA and KdpB subunit, thus creating a "chimeric K⁺ uptake system." In the authors' proposal, the role of KdpA is downgraded to contribute only passively as a quasi-immobile subunit that contributes to ion transport with an open half channel on the luminal side and with its selectivity filter guarding the entrance to a "horizontal" tunnel. This tunnel connects the selectivity filter in KdpA with the "conserved canonical binding site of P-type ATPases" in KdpB and allows propagation of K⁺ to this binding site. After a subsequent conformation transition the horizontal tunnel collapses at the interface between KdpA and KdpB (thus producing transiently an occluded state). In addition, a movement of TM2 and TM4 of KdpB opens another half channel (or tunnel) allowing the release of K⁺ from its binding site to the cytoplasm. This concept is insofar exciting as it brings back again the active transport mechanism to KdpB only, the P-type ATPase subunit, and thus moves this K⁺-pump complex closer to the other relatives of the "family".

"State 1" as derived from the EM images was described as: "it resembles a late nucleotide-bound E1 conformation." Characteristic properties were that (1) the A domain was rotated away from the N and P domains and the N and A domains were clearly separated, (2) a tunnel connected the selectivity filter in KdpA with the conserved canonical binding site around Pro264 in KdpB, (3) three electron densities were identified that could be assigned to potassium ions, one at the outer S1 position of the selectivity filter and two coordinated within the inter-subunit tunnel, but not in the canonical site, and (4) no tunnel was found between the conserved binding site and cytoplasm. Two more findings were reported that will be discussed below: (1) "we can find an additional weak density in the cryo-EM map that we assigned to an AMPPCP molecule" (p.4), and (2) "State 1 is an only partially K⁺-loaded late E1 state ... (p.8)."

"State 2" was characterized as: "it resembles an E2-P conformation," and "represents a late E2-P state after ion release, in which the inward-open tunnel is already partially collapsed." Important features were: (1) the A domain formed a tight interface with the N and P domains, and the TGES motif of the A domain was in place to dephosphorylate Asp307, (2) the salt bridges were broken due to rearrangements of the P domain, and the subunits KdpA and KdpB were well separated, (3) most likely AlF₄⁻ was coordinated by a Mg²⁺ instead of AMPPCP, of which no density was seen, (4) the inter-subunit channel ended at the subunit interface between KdpA and KdpB which disabled any K⁺ transfer from KdpA to the binding site in KdpB, (5) TM2 and TM4 of KdpB were moved apart by 5 and 15 degrees, and opened a tunnel reaching from the K⁺ binding site to the cytoplasm.

Based on these findings a Post-Albers type transport cycle was proposed (Figure 6) with the anomaly that the entrance for ions into the pump is reversed to the other P-type ATPases, i.e.

open to the extracellular (= luminal) side in the E1 conformation, and open to the cytoplasm in the E2P conformation. (This suggestion was first introduced by Haupt et al. (2004) J.Mol.Biol. 342:1547.)

The cryo-EM experiments and the control EPR studies were performed accurately and data analysis was conducted with state of the art techniques. The work was presented in a clear manner and is well comprehensible. Data presentation and visualization are fine, and the substantial supporting material is very helpful and appreciated.

There are, however, two major issues that need detailed consideration.

(1) Both states of the KdpFABC complex selected for detailed analysis were introduced as "a late nucleotide-bound E1 conformation" (state 1) and "a late E2-P state after ion release" (state 2). The limitation on only two states is quite arbitrary. They appeared simultaneously in the same buffer (K^+ , Mg^{2+} , AMPPNP, AIF₄⁻) and an equilibrium distribution between states may be assumed, before they were shock-frosted. According to the underlying Post-Albers scheme it is obvious that both states cannot be adjacent in the transport cycle. A state without AMPPNP and AIF₄⁻ as intermediate is the minimum requirement. The "invisibility" of such a state could be possibly explained by the high concentration of the so-called inhibitors which would deplete the interjacent plain E1 or E2 state. However, the statement on p.4 with respect to state 1, "we can find an additional weak density in the cryo-EM map that we assigned to an AMPPCP molecule," raises an important question. The authors decided (quite arbitrarily) to pool all particle images with an appropriate arrangement of the N, P, and A domains as a single state. Is it possible that this choice produced a questionable average structure? When a distinction between particles with and without a density "assigned to an AMPPCP molecule" would be made, two different states may be revealed. In addition, a clarifying explanation to the "only partially K^+ -loaded late E1 state" may be found therefrom.

(2) Concerning the proposed Post-Albers scheme (Figure 6), it has to be stated that it contravenes the functional studies from Siebers and Altendorf (J.Biol.Chem., 1989, 264:5831), in which the KdpFABC complex was maximally phosphorylated by [γ -³²P]ATP in the absence of K^+ , and addition of K^+ induced dephosphorylation. They explained their extensive and consistent set of data with the "standard" Post-Albers scheme, i.e. K^+ induced dephosphorylation but phosphorylation by ATP without K^+ . Until today no functional study has been published that challenged their findings (and interpretation), although the few papers with a different mechanistic assumption just ignored these findings. The weight of structural insights from "inhibited" states of the KdpFABC complex that "resemble" functional states, as presented here, should not be deemed high enough to brush results under the carpet that represent pump function under (almost) physiological conditions. In addition, according to the proposed new transport mechanism, a "late" E1 state that precedes directly enzyme phosphorylation would require an occupation of the canonical site by a K^+ . A critical discussion of the current discrepancy of functional and the presented structural findings is the least the authors should provide. Maybe, the statement on p. 7 "the presented structures finally allow to propose a conclusive mechanism for active transport of K^+ via KdpFABC" is still somehow premature.

Reviewer #2:

Remarks to the Author:

The manuscript entitled "Cryo-EM structures of KdpFABC reveal K^+ transport via two inter-subunit half-channels" describes two structures of the membrane protein complex KdpFABC in two different conformations, leading to the proposal of a novel mechanism for K^+ transport.

The work presented appears to have been well conducted. The study is important as it reveals molecular features of the mechanism of a K⁺ pump that is unique to bacterial systems and that plays an important role in the physiology of many bacteria. Strikingly, the architecture of the complex and associated mechanism are also unusual since they display features found both in K⁺ channels and P-type ATPase pumps. The study will be of great interest to structural biologists and microbiologists in general and to researchers working on the mechanism of ion transport in particular.

I have a few problems with the manuscript.

1- The assignment of a catalytic states according to the Post-Albers cycle (state 1 according to the authors corresponds to the E1 conformation and state 2 to E2-P conformation) is an important aspect of the analysis of the structures and it should be better supported by a Figure showing a more careful comparison with existing structures of other P-type ATPases. This new figure will also help the reader that is less familiar with the intricacies of the ATPase ion pumps. In this context, the authors refer as an important aspect of their assignment the position of the TGES motif in the A domain but this motif is not clearly indicated in the figures.

2- The assignment of the catalytic state also involves as discussion of the possible presence of nucleotide. The authors use as arguments to support this presence the existence of weak density associated to the N domain and the comparison of one of the KdpFABC structures with a structure of SERCA in supplemental Figure 5.

The weak density for the nucleotide is very difficult to evaluate as presented in Supp. Figure 4d. I suggest the generation of a stereo-image to better present this data. The same applies to the density for the AlF₄⁻ molecule in panel e) of the same figure.

The superposition of the nucleotide binding sites of KdpFABC and SERCA in Supplemental Figure 5c) is supposed to support the idea that they are similar. However, the superposition clearly shows that the two conformations are different and the sites do not match so it is unclear how the conclusion is reached. See further comments on these figures below.

3- The reasoning behind the analysis of the SPR results is a bit confusing since the authors state "However, it [AMPPNP] stabilized an E1 conformation with a narrow distance distribution between the spin-labeled residues centered at 4nm resembling the observed distances in state 1...". As far as I know a P-type pump can bind ATP (or AMPPNP) both in E1 and E2 states. It is therefore not obvious the reason why the authors have assigned the state E1 to the conformation in solution by SPR.

4- The authors should explain in more detail the rationale for assigning a K⁺ to the densities present in the tunnels observed between KdpA and KdpB? Could these densities be well ordered water molecules that are detectable even at this low resolution?

5- In page 6 the authors speculate that water molecules oriented by a second layer of the selectivity filter (Asn114, Gly232 and Gly345) also coordinate the K⁺. What is the basis for this speculation? Since this statement does not seem to be based on any data provided by the structure it seems that this level of speculation is unnecessary.

6- In page 6, the authors state " suggesting the possibility of a coordinated movement of partially hydrated K⁺ ...". I am not sure what "coordinated movement" means.

A major problem I have found with the manuscript is the organization and quality of the figures that do not facilitate the analysis and appreciation of the work. In particular, the authors have opted to use a weak and strong shade of the same color in the two different structures. This choice instead of helping the visualization of the different conformations makes it harder.

Figure 1: panel c) is very difficult to understand since I cannot distinguish the different domains

from the two different structures. In addition, the authors refer to Figure 1c to indicate the differences in the relative position of the spin labels but the corresponding spheres are barely visible. I suggest that the authors use the same color scheme for the two states and simply show the two states side-by-side. Additionally the authors have used superpositions in several panels and in general these are very confusing, for example in Figure 1 panels a) and b) just show a mash of densities and helices which have little use. I strongly suggest that the authors avoid the use of superpositions (except for a few cases and in these more care should be taken to make the figure useful) and just place the two states side-side and in the same orientation.

Figure 2e: (and in other representations of state2, see below) the actuator domain (light yellow) is barely visible.

Figure 4: indicate TM2 and TM4 in panels a) and b) so that they are more easily related to panels c) and d) and better support the point made in the main-text.

Supplemental figure 3. In panels 3b, 3c, 3f and 3i the actuator domains are barely visible. Since the structures are presented in different panels better use the same exact color scheme across all structures to better compare positions of domains. Superposition in panel c) is not useful since it is just a mash of helices.

The authors also show representations that are related to each other in different figures, making it harder to understand the points raised in the text. In particular, panels that present phosphorylation at S162, the binding site of nucleotide and the AIF4- binding site are spread between Supplemental Figures 4 and 5. It would be much more reader-friendly if these panels were put together in a single figure, leaving the density for the secondary structure elements shown in Supplemental Figures 4a and 4b as a separate figure, ideally with a larger size for better analysis of density quality.

I also found that the main text presents a few awkward sentences that I strongly suggest should be changed:

page 2, 1st paragraph in main text

"... the highly K⁺-affine, actively pumping P-type ATPase...". Needs to be rewritten.

"...the KdpFABC complex is the only known chimera composed of 4 different subunits". Besides the unconventional use of the term "chimera", the sentence appears to state that KdpFABC is the only known "special" complex composed by 4 different subunits. I am sure that this is not what the authors intend to say and it should therefore be changed.

Page 5, 2nd line:

" ... the phosphorylation does not lead to the initially proposed auto-inhibited...".

I suggest something more like "... phosphorylation at S162 in different conformational states of KdpFABC does not fit with the previously proposed model where phosphorylated S162 was thought to lock the A and N domains in an auto-inhibited conformation."

Page 7, 1st paragraph:

"...composed of the two half channel is promoted by the absence of any of the previously suggested conformational changes within KdpA and KdpC."

Probably say that "...composed of two half channels does not involve the previously suggested conformational changes...".

Page 7 2nd paragraph:

"After decades of studies, the presented structures finally allow to propose a conclusive mechanism for transport of K⁺...".

This rather grandiose sentence would greatly improve if the word "conclusive" was omitted.

Whether this important study is the defining work on the mechanism of KdpFABC transport mechanism will be determined in the future with other studies.

Page 7 2nd paragraph:

“Other than expected, not solely the channel-like subunit B facilitates K⁺ translocation, but rather the combination of two joined half-channels ...”. This sentence could be simplified.

Reviewer #3:

Remarks to the Author:

KdpFABC is an important K⁺ import system helping bacterial cells deal with several environmental stressors. A crystal structure of the entire complex (actually a mutation, R116, with decreased activity) has been determined and published last year (Huang et al., Nature 2017). In the present manuscript, the authors present two cryo-EM structures of the KdpFABC complex, which they link to two distinct states of the transport cycle. The cryo-EM analysis appears straightforward and robust and the work is supported by ATPase assays along with EPR and MS analyses. The novelty of the current results rests on the proposal of a new mechanism for cytosolic K⁺ release, suggested previously (Huang et al, 2017) to occur through the channel (KdpA) pore. In the present work the authors suggest that the release instead occurs through a newly observed tunnel in KdpB leading from the canonical cation binding site in KdpA to the cytoplasm and forming upon transition to the proposed E2P state. The Stokes group (Huang et al, 2017) had previously proposed that occlusion of substrate binding sites occurs through a rearrangement of KdpC at the periplasmic side and by conformational changes of the so-called coupling helix/loop of KdpA at the cytoplasmic side. In contrast, here the authors promote the idea of occlusion of the K⁺ binding sites in KdpA from K⁺ ions in KdpB by a closing of the previously observed horizontal tunnel connecting the two interacting proteins. The authors further propose that unlike prototypical K⁺ channels, the KdpA channel remains closed as no conformational changes of the proposed gating elements are observed, and that K⁺ instead is shuttled through the open inter-subunit tunnel to the canonical substrate-binding site in KdpB by a ‘knock-on’ mechanism. If true, this would represent an exciting, albeit unusual finding. The validity and general acceptance of this proposal would be greatly aided by a transport assay some evidence directly supportive (in particular showing transport of K⁺ without an opening of the cytosolic gate of KdpA). Without such a demonstration, the conclusions remain very speculative. Unfortunately, the presentation of the paper is at times confusing and discrepancies between the current structures and previously determined crystal structures need to be addressed. The description of some of the methodology is also confusing.

Specific points:

1. The proposed mechanism stipulates that the channel (KdpA) remains closed throughout the transport cycle. There needs to be some experimental demonstration of this to support such a conclusion. The authors should consider (disulfide) cross-linking to immobilize pore linking helices and/or the gating helix to prevent channel opening and to confirm that transport activity is maintained when channel opening is prevented.
2. The physiological relevance of the simultaneous use of non-hydrolysable ATP analog and the Pi mimic is questionable. Why was ADP excluded and what happens (functionally) in the presence of ADP/ALF4-?
3. The continuous tunnel (along the selectivity filter and horizontal tunnel between KdpA and kdpB) was observed in the previous crystal structure (after exclusion of the mutant R116 sidechain, which mimics a K⁺ ion at S1). The assignment of K⁺ ions to any density in the current work is speculative and figures depicting bound K⁺ ions (e.g. Figs 2b-d, g), made to look like Fo-

Fc difference maps, could be misleading in the absence of expanded figures showing density for all surrounding ligands (basically, it could be noisy density made to look like specifically bound elements based on the density threshold and carve around the placed ion). The same is true for proposed nucleotide /ALF4- density, although the authors explicitly state that the density is weak. The EM density evidence is therefore currently insufficiently demonstrated.

4. The authors should consider focused refinement of subdomains/domain pairs in an effort to improve the resolution to clarify binding of nucleotides/ALF4-. As it stands, the placement of nucleotides/AIF4- is highly speculative with what appears to be very noisy density in the relevant regions.

5. The crystal structure of the complex excluded K⁺ binding at the canonical site, which was instead proposed to be occupied by a water molecule. The currently proposed mechanism necessitates K⁺ binding at this site, which would lead to introduction of a positive charge next to K586. The authors should discuss this discrepancy in greater detail (as mentioned in point 2, assignment of density within the tunnel as K⁺ ions is speculative).

6. In the current scheme, it is unclear how K⁺ would be released against a concentration gradient from the E2-P state considering knock-on (invoked here as the primary mechanism driving K⁺ binding and transport) is prevented by the occlusion of the canonical site from ions bound in KdpA. The authors should explicitly present a superpositioning of the relevant KdpB residues in both states.

7. The entire purification of EM samples is done in the absence of K⁺, which is introduced in combination with the inhibitors prior to grid preparation at a lower concentration than the Na⁺ present in the purification buffer. In the absence of K⁺, many K⁺ channels have been known to conduct Na⁺. Can the authors rule out Na⁺ specific effects based on ATPase stimulation (or lack thereof) for the wild type protein used here at different K⁺/Na⁺ ratios?

8. The authors conclude that KdpC likely plays only a stabilizing role. How likely is this considering the extensive hydrogen bonding contacts between KdpA and KdpC, which would likely couple any changes in the channel domain to conformational changes in KdpC as previously proposed based on the crystal structure (Huang et al, 2017)? This should be discussed further in terms of known KdpC mutations that alter transport and ATPase activity.

Reviewer #4:

Remarks to the Author:

Stock and colleagues present two cryo-EM structures of the prokaryotic P-type ATPase, KdpFABC determined in the presence of AMPPCP and AIF4- at resolutions of 3.7 and 4.0 Å. Comparison of these structures with an existing crystal structure of KdpFABC reveals that these structures represent novel states for the transporter. Comparisons with structures of other P-type ATPases suggest that state 1 corresponds to an E1 conformation while state 2 corresponds to an E2-P conformation. ATPase assays and EPR distance distributions, which demonstrate that AMPPCP and AIF4- inhibit ATPase activity and stabilize the transporter in E1 and E2-P states, provide additional evidence for the assignments. In the cryo-EM density map, the authors note the presence of densities that they have assigned as ordered K⁺ ions. The presence of these density peaks along with measurements of a previously described cavity extending between KdpA and KdpB allow the authors to delineate a putative ion translocation pathway that extends from the selectivity filter region of KdpA through KdpB into the cytoplasm. Based on the novel ion conduction pathway and on conformational changes in the P domain of KdpB subunit, the authors present a transport mechanism where K⁺ ions are concentrated by a high affinity site in the KdpA selectivity filter, moved through the pathway towards KdpB and then actively transported through KdpB via an

alternative access mechanism.

The authors conclusions are well supported by multiple lines of evidence. However, the presentation of the data makes it somewhat difficult to follow the authors' argument for their transport mechanism.

Several changes to the manuscript could be made to improve the presentation of the data.

1. The authors title Figure 1 as an "Overview of KdpFABC cryo-EM structures in different conformations". However, the figure is lacking a panel presenting an overview of the structure of KdpFABC to provide readers not accustomed to analyzing P-type ATPase structures a frame of reference for understanding the different conformations displayed by the two structures and for the detailed analyses presented in the subsequent figures.
2. Continuing from point 1, maintaining a single color scheme throughout the manuscript would aid readers in understanding how movements of the transporter during explain the transport mechanism.
3. In Figure 1c, it is nearly impossible to identify the spheres used for the spin labels in the image. Please use an alternative color or highlight the spheres in another manner.
4. There are several statements in the manuscript that are not clear. For example, it is written on page 5 "Hence, the EPR data demonstrate that AMPPCP is required to achieve state 2, although it is not visible in the cryo-EM map. Notably, also identical structures of SERCA in an E2-P conformation with [3B9R] and without [3N5K] AMPPCP bound were solved".

Comments:

1. The authors should more fully address the mechanisms by which nucleotide state determines that state of the transporter and how changes in nucleotide state drive the reaction. Also, why is AMPPCP required for stabilization of the transporter in state 2, yet it is not present in the structure?
2. In Figure 2, the authors present density peaks that they have attributed to K⁺ ions. While the peak in panel b appears to be partially coordinated by Q116, N239 and G468, the peaks in the remaining panels are poorly coordinated or not coordinated at all. At the intermediate resolutions of these reconstructions, how do the authors justify that these density peaks correspond to K⁺ ions?
3. The authors suggest that the KdpC subunit serves to increase the K⁺ affinity of the transporter in a mechanism similar to the β subunits of the Na⁺, K⁺ ATPase. Such an argument would be strengthened by an analysis of the electrostatic surface potential of the opening of the transporter or by functional assays demonstrating that mutation of specific residues in KdpC inhibits K⁺ uptake against large K⁺ concentration gradients.

Reviewer #5:

Remarks to the Author:

First of all I would like to congratulate the authors on a beautiful piece of research revealing two novel conformational states (~ 3.7 Å and ~ 4.0 Å) of the potassium transporting Kdp complex using cryo-EM. Both states are key intermediates of an active transport cycle that has been the focus of much speculation previously in the literature and which we get a glimpse at now for the first time.

The Kdp system is an highly interesting K-transport system since it combines two very distinct super-families of transport proteins with completely disparate modes of action. Namely, the SKT channels and the P-type ATPases. The fact that Kdp only function when both of these distinct components are together present a unique and fascinating case study in evolution, and much can be learned about both classes of proteins from this obligate complex and its mechanistic function. This is well exemplified in the presented manuscript that bring to life completely new ideas and suggestions on how both channels and pumps might function mechanistically, that would not have been thought of if one focused solely on one type of transport.

There can be no doubt in my mind that the work and the topic are highly relevant and appropriate for the Journal, and overall, this reviewer strongly recommends publication in Nature Communications.

However, there are multiple things that I would like to see addressed before publication. Some are related to the method and must be addressed prior to publication, and some are related to the analysis and conclusions drawn from the analysis and should preferably be addressed, or at least better explained in the manuscript. I do not think that any follow up experiments are needed for now.

As a short insert, line numbers would have been nice, and allowed more detailed comments on actual wording.

A few textual/figure notes:

Page 2 bottom (line 5 of text): KdpFABC is not a P-type ATPase, KdpB is. Please make sure to refer to KdpFABC as a complex, not a P-type ATPase. This language mix-up is common in the Kdp literature, but should be avoided for clarity. In a similar vein KdpFABC would also not be called an SKT protein.

Page 2: The SKT super-family name should be introduced with KrtB and TrkH in the beginning of the manuscript to avoid confusion, when KdpFABC is introduced, since KdpA is also member of the same group of proteins with a similar physiological function (potassium homeostasis).

Page 3 top: When the gating loop is introduced it should be stated that it is found in the D3 repeat.

Page 8 middle: "... as proposed 30 years ago.." Yes, but for Na/K ATPase, and H/K ATPase, not for KdpC. Please clarify this sentence. Btw, why is it relevant to know that this was proposed 30 years ago?

There are no clear figures on the domain movements of the ATPase between the states. Only an overlay in fig. 1 and an overview in supp. fig 3.. This should be amended. It is very hard currently to visualize the P domain movement and the position of the TGES loop.

Figures are generally very confusing. They need more labels, a bare minimum would be to label the TM helices (all figures, but grievous examples would be fig. 2b-d, f-g and fig. 4).

Fig 1 in particular is very confusing. The authors should separate the states to make it easier to see them, and remember that many people are colorblind. The two colors for the states are very hard to distinguish (I showed that figure and Supp fig 6 to a color blind colleague, and he could not see that there were two states, not identify the red line in the panels of Supp fig 6).

There are 3 main figures showing the tunnels (Fig 2-4). This is overly redundant. I suggest the authors merge figure 3+4, and move parts of them to supplementary.

The point of figure 5 and that one sentence on page 8, that KdpC could function like the beta subunit of the Na/K ATPase, is extremely speculative and irrelevant for the rest of the paper. It should be either deleted or moved to supplement.

In fig 6 you should highlight the two states you have models for. In Fig 2 you should explain what the red line is for (1.4 Å for water or for K ion radius?).

Supp. Fig 4 is the best figure of the manuscript in my opinion.

--

On the methods:

Cryo-EM: Generally the cryo-EM work is well described and everything seems to have been done correctly as far as I can tell from the description. However a few things need to be clarified. In Supp. Fig 1d, the flowchart need to explain where the initial reference state 1 and state 2 came from for the first 3D classification. It should be noted how many ab initio models were used or if at all.

It is unclear when the Kdp crystal structure was used in the process. No mention is made of a low pass filtering of the input model, but I assume that this was done?

Why did the authors use two different B-factors for sharpening and for the generation of images. E.g. State 1 uses -154 \AA^2 but -205 \AA^2 for the figures and manual inspection. The rationale should be explained.

Supp fig 2c: the A and N domain appear to be missing? What happened to the contour level here?

The authors show the sample to be phosphorylated at S162 in the two observed cryo-EM states. This is important for the downstream analysis. In the crystal structure paper (Huang et al Nature 2017) it is demonstrated that the phosphorylated sample has no ATPase activity, and the observed activity is therefore proposed to arise from a fraction of the purified sample that is not phosphorylated at S162.

In the manuscript the MS data (supp table 2) seems to show the same pattern. Namely that a fraction of the purified sample is not phosphorylated. I assume that this fraction is what gives the observed ATPase activity in supp table 3. If so, this should be explained, or alternatively, if the authors truly believe that the Kdp complex can exhibit ATPase activity while being in the S162 phosphorylated form this should explicitly be stated, and then backed up by some evidence. That scenario would be hard to imagine, given what we know from SERCA and other P-type ATPases about the function of the TGES loop, and does not fit the data from the Huang et al paper.

I tentatively conclude that the states observed in the EM maps are phosphorylated and inhibited states. It would be very interesting to hear the authors view on this aspect of their data, and furthermore it should be much better explained in the manuscript that the states observed are inhibited states, if the authors believe so.

It would also be interesting to know why the authors decided to use an inhibitor mix of AIF4 and AMPPCP for the work. I assume this was based on the 3B9R structure of SERCA?

ESP:

The ESP experiments are described in a somewhat confusing manner. Generally the experiments does not really contribute to the story, and should be re-delegated to supplementary material (move panel fig. 1d to supplementary).

First a discrepancy: In Fig. 1d the E1 grayed out area is centered on 4 nm, and does not match the red curve in supp fig 6d (with 5 mM AMPPCP) which is centered on 3 nm. How is the grey area denoted E1 calculated?

As I understand it from especially Supp fig. 6, the observed state 1 and state 2 are used to calculate the red predicted distance distributions in panel d of supp. fig. 6, but in fig. 1 the same prediction is then used to denote the gray areas as E1 and E2-P instead of state 1 and state 2. This is almost circular logic. The ESP experiment does only show that AIF4 is needed to reach state 2 in the sample, not that state 2 is the E2-P state.

Calling it E2-P in Fig. 1d (while it may be true) gives the impression that the E2-P conformational conclusion is based on more than the structural comparison of state 2 to SERCA crystal structures. Thus these labels for the gray areas in fig. 1d should be something like 'theoretical/pure state 1' 'theoretical/pure state 2' or similar.

One speculates where in this spectrum the Kdp crystal structure would be. A quick measurement reveals that in the crystal model the relevant distance is ~3 nm. Right between the distances of 4 nm for state 1 and 2 nm for state 2. The crystal structure was solved from a sample with 5 mM AMPPCP, and so if measured would presumably show a trace similar to the orange trace in fig 1. However the actual crystal structure did not contain AMPPCP in the binding site. It would be informative to see the theoretical curve for the crystal model as well in supp. fig 6d.

Supp fig 6 and fig 1: Out of interest, why does the x axis start at 1.5 nm?

--

On the analysis:

All of this bring us to THE key analysis. What are the two observed states?

The authors believe that they are observing E1 and E2-P. This is likely correct (but arguably an S162(P) inhibited E1 and E2-P). However the ESP experiments does not prove this as argued above, and the actual EM density for the inhibitors is less than convincing (Supp fig. 3d and e). I am somewhat shocked that it is apparently acceptable in the EM world to model small molecules into density of this quality, but I am glad the authors show the density so it is possible to evaluate. This should serve as inspiration to other cryo-EM papers. According to Supp Fig 2 c and d the actual resolution in the area of the maps are ~6 Å so the lack of definition is not surprising.

In the end, the only real argument that the observed states are E1 and E2-P is the superposition of the states with SERCA E1 and E2-P, as far as I can tell. And this will have to be done in the cytosolic domains of KdpB, which are the parts of the EM maps that are the least well defined (Supp fig 2c+d). Thus is unacceptable that this superposition is not shown in any figure, nor is the details of how the superposition was done explained, or the RMSD value of the superposition reported.

The only superposition figure is a detail of the AMPPCP site found in supp fig 5c, and there the overlap actually looks pretty bad.

Along the same line, it is very unclear in the manuscript which SERCA model was used for the comparison to denote conformational state. Only in the methods section are the pdb-id's 1T5S (SERCA E1-AMPPCP) and 3B9R (ERCA E2-P) mentioned, but here they are used to guide the initial model building of AMPPCP and AIF4 only? I assume (since it is not explained) that these pdb-id's are then also used later to conclude on the conformation state?

How many other conformational states of SERCA or other P-type ATPases did the authors try before concluding that 1T5S and 3B9R were the best fit?

If 3B9R is the model we are comparing to for state 2, the following comes to mind. In 3B9R the E183 from the TGES loop is 2.4 Å from the coordinating water that will coordinate the AIF4 (it is a transition state analogue), and is key for the coordination. What is the distance of the equivalent residue (E161) in state 2? This is also related the the phosphorylation observed on S162. Does the TGES loop actually have the same position related to the AIF4 as in SERCA E2-P transition state, or do we have an intermediate state where the AIF4 is bound but not really mimicking the E2-P transition state? How sure can the authors be, if there is ~6 Å resolution in this area? How does the model fit the density here?

More figures and some RMSD numbers would help to understand how well state1/2 fits E1/E2-P as expected from homology to SERCA.

The whole analysis is based on the correct identification of the states so this is a key analytic step, and it should be much better explained in text and figures how the conclusion of E1 and E2-P was reached!

In the end, I do think that E1 and E2-P are likely correct, but it took some time for me to get there due to the lack of clarity in the manuscript in this part.

In particular I need a figure of the TGES loop and its location in relation to Asp307 of the P domain

in both states and the matching map density. This is very hard to see from the current figures (except supp fig 5b, a panel that should be part of the main figures).

The tracing of the tunnel or tunnels then become the main focus of the rest of the manuscript. Generally speaking there is just too many figures of the tunnel features. I strongly recommend the authors merge at least figure 3 and 4, and use the extra space to show show of the other features described above.

Figure 2 show the exciting observation of some spherical density in the tunnel. These are modeled as potassium ions and naturally are of particular interest.

It is not clear to me that a potassium ion would give a stronger peak than carbon or oxygen (eg water) in an EM potential map, in particular a map that has been blurred. This is not crystallography where the signal (sigma level) is proportional to the number of electrons. This made me curious, and I looked into the literature on the topic, and found a very interesting paper from Wang et al. *IUCrJ* 5 (2018) "Identification of ions in experimental electrostatic potential maps" (<https://doi.org/10.1107/S2052252518006292>).

I wonder if the authors could use the techniques described in this paper (of blurring the maps) to help strengthen their argument that the observed peaks are indeed a positively charged potassium and not eg. water or part of a larger molecule such as lipid or detergent?

I also wonder if the fig 2b-d peaks have been cropped so that the surrounding density is not visible? This is not explained in the figure text.

For now, the modeling of these 3 peaks as specific atom types should be done very carefully and with great respect to the surrounding chemical environment. Since this again is a key aspect of the analysis much better figures should be presented. Fig. 2 Panels b-d are not sufficient to properly evaluate the chemical environment of these peaks from the readers perspective.

Site 1 (fig. 2b) and site 2 (fig. 2c and fig. 2g) seems to be ok, given the chemical environment. Both appear to be in or close to the selectivity filter of KdpA, which is known to bind potassium, and both spheres are coordinating to residues that might stabilize a positive charge.

However site 3 (fig. 2d) appears extremely unlikely as it is presented now. Apparently this positive ion is coordinated by the two apolar residues Ala227 and Leu228 and nothing else? I would like to see a much clearer picture of this site, and a discussion of the chemical environment of this last ion. It seems impossible to me that an ion would be fixed in this position in about ~220k particles (of ~535k total) without any kind of chemical interaction to help keep it there?

Furthermore the energy cost of burying a positive charge here in the middle of the membrane without clear coordination needs to be discussed. This would seem to me to have a very high energy cost, given our knowledge from potassium channels where previous work has shown that the K⁺ needs to be hydrated or in a very specific selectivity filter.

It would be interesting to calculate the cost of this burial, using e.g. MD. As the radius of a hydrated K⁺ ion does not fit with the tunnel, it must be at least partially dehydrated in the suggested position.

The cost of burying a single positively charged ion in the membrane is theoretically in the order of 40 kcal/mol (Honig et al, *ann rev biophys biophys chem* 15 (1986))(for reference, ATP hydrolysis releases ~12 kcal/mol in a normal cell).

Since the tunnel can create some partial hydration, the cost will be less, but still significant. The authors suggest that the tunnel and the filter are at all times loaded with potassium and ready for the next round of transport so to speak, meaning that the tunnel would presumably always have at least one buried ion within it. The energetic strain on the system for this to be correct seems to make this part of the suggested model highly unlikely. The authors should discuss this aspect of their model.

I am sure the authors already know this, but for the sake of the argument, I would like to point out the the refinement done in phenix.real_space_refinement does not include an electrostatic

term

to its target function, and therefore the refinement does not take into account the specific charge of any residues or ions. Only geometrical parameters (bond lengths, angles etc) are optimized. Thus the model refinement does not test if the chemical environment of the site 3 (fig. 2d) K⁺ ion is a stable one. Refinement of the model using MDFF, with explicit charges of the potassium comes to mind, as this might say something about the stability of an ion in this unusual position. Have the authors considered trying this?

Another related thing that should be addressed is the observation that in the EM models the 'canonical' KdpB site next to M4 is apparently empty. In the crystal structure a peak was observed here, modeled as a water. Based on this model and analysis this peak must likely be potassium instead. So, why do the authors think there is no K⁺ in this position if state 1 follows the crystal conformational state (as per page 8 discussion). This discrepancy should be discussed. More generally the authors need to address why they apparently observe a K⁺ ion in a very chemically unfavorable site, and no K⁺ ion in the adjacent much more favorable 'canonical' site of KdpB? To help the argumentation, a figure with density of this particular area would be useful. This could be part of fig. 2.

In state 2 the authors observe a collapse for the tunnel directly related to the breakage of the coupling helix to the P domain, and the very static KdpA structure (fig. 5a). This is the major finding of state 2, and very interesting. Furthermore, the exit tunnel/pathway observed in state 2 is not the classical entry/exit pathway known from other P-type ATPases, but a completely new path from the M4 site to the cytosol. This interesting aspect of the model should be addressed more clearly and the figure improved. It is currently very hard to follow clearly the exit tunnel in the figures. It is currently unclear how the ATPase architecture that couples ATP hydrolysis to gate opening/closing could have changed so dramatically from other P-type ATPases, and this should be discussed.

Finally, I would like to argue that the manuscript does not do full justice to the copious previous literature in the field. The biochemical literature in the field needs to be referenced more, since the proposed model does not really agree with most earlier biochemical studies as far as I can tell.

Some of the literature that the manuscript would benefit from discussing include:

- Bramkamp et al. *Biochem* 44 (2005): Mutation of residues 583 and 586 of KdpB creates uncoupling of ATPase activity and transport. This indicates that these residues that are in the tunnel are part of the coupling from binding site to ATP hydrolysis site, but not part of the translocation pathway. How does this fit the presented model?

- Becker et al *Biochem* 46 (2007): Same point as above examined in more detail.

Bertrand et al *J bacteriol* 186 (2004): Mutations in KdpA change the cation affinity of the complex. Does this mean that the proposed 'canonical' site of KdpB does not distinguish between ions? All other P-types have extremely strong selectivity in the canonical site. The authors should discuss why they believe the KdpB does not need such a feature.

- Buurman *JBC* 270 (1995): They screened all Kdp subunits and identified mutations that affect substrate specificity. These were all clustered in KdpA.

- Dorus et al *JBC* 276 (2001): They saw the same thing, but focused on KdpA. Similar point as above. For these 3 papers above on selectivity, try to explain the following: In an active transporter substrate selection must be happening in the binding site that becomes occluded to ensure tight coupling. A KdpA 'prefilter' that allows both K⁺ and water to pass would not work, because then water would be enough to trigger ATP hydrolysis when bound to the KdpB binding site.

For any highly coupled active transport (like Kdp postassium transport), only when the correct

substrate is bound can occlusion and transport take place. This is a fundamental principle of coupling in P-type ATPases and generally for active transport (reviewed numerous times). Do the authors think that Kdp has a different mechanism to achieve coupling, and if so, what could it be?

- Puppe et al. Mol Microbiol 6 (1992): They showed the KdpB Asp300 mutant increased ATPase activity and decreased K⁺ ion affinity, signs of uncoupling. This residue is part of the link to the D3 'coupling' helix in the Huang et al paper. Explain this observation using the proposed model. In this model the coupling helix is static and breaking this bond thus might free the ATPase to hydrolyze ATP faster, but why would K⁺ affinity decrease and uncoupling occur?

In conclusion, I have a lot of questions and some concerns, especially on the proposed model, but in the end, this is for the authors and others to challenge with future research, and will set the stage for a lot of interesting discussions in the field. Overall the methodology is sound and the findings highly interesting, and I would again recommend publication after some edits and clarifications. I am looking forward to discussing these fascinating results with the authors at future conferences. Congratulations again!

Reviewers' comments/authors' replay:

Reviewer #1 (Remarks to the Author):

In this interesting and important contribution cryo-electron microscopy was used to elucidate the structure of the KdpFABC complex. Two new structures with a resolution of 3.7 Å and 4.0 Å could be presented. In contrast to the previous single structure of the complex with almost atomic resolution that was gained by X-ray diffraction analysis from protein crystals, the cryo-EM structures are obtained from single molecules, solubilized in DDM, which did not need to make possibly distorting crystal contacts.

The analyzed samples contained KdpFABC complexes in the presence of saturating concentrations of KCl, MgCl₂, AMPPNP, and AlF₄⁻. These substrates, partly indicated as 'inhibitors', were able to bind to the complex in different combinations and induced diverse states. During the analysis procedure the authors confined themselves to focusing on two states, named 'state 1' and 'state 2'.

In contrast to assumptions of the ion transport mechanism presented in previous papers, now solid evidence is presented that the pathway of the ions is not restricted to the K⁺-channel like KdpA subunit but leads through both the KdpA and KdpB subunit, thus creating a "chimeric K⁺ uptake system." In the authors' proposal, the role of KdpA is downgraded to contribute only passively as a quasi-immobile subunit that contributes to ion transport with an open half channel on the luminal side and with its selectivity filter guarding the entrance to a "horizontal" tunnel. This tunnel connects the selectivity filter in KdpA with the "conserved canonical binding site of P-type ATPases" in KdpB and allows propagation of K⁺ to this binding site. After a subsequent conformation transition the horizontal tunnel collapses at the interface between KdpA and KdpB (thus producing transiently an occluded state). In addition, a movement of TM2 and TM4 of KdpB opens another half channel (or tunnel) allowing the release of K⁺ from its binding site to the cytoplasm. This concept is insofar exciting as it brings back again the active transport mechanism to KdpB only, the P-type ATPase subunit, and thus moves this K⁺-pump complex closer to the other relatives of the "family".

"State 1" as derived from the EM images was described as: "it resembles a late nucleotide-bound E1 conformation." Characteristic properties were that (1) the A domain was rotated away from the N and P domains and the N and A domains were clearly separated, (2) a tunnel connected the selectivity filter in KdpA with the conserved canonical binding site around Pro264 in KdpB, (3) three electron densities were identified that could be assigned to potassium ions, one at the outer S1 position of the selectivity filter and two coordinated within the inter-subunit tunnel, but not in the canonical site, and (4) no tunnel was found between the conserved binding site and cytoplasm. Two more findings were reported that will be discussed below: (1) "we can find an additional weak density in the cryo-EM map that we assigned to an AMPPCP molecule" (p.4), and (2) "State 1 is an only partially K⁺-loaded late E1 state ... (p.8)."

"State 2" was characterized as: "it resembles an E2-P conformation," and "represents a late E2-P state after ion release, in which the inward-open tunnel is already partially collapsed." Important features were: (1) the A domain formed a tight interface with the N and P domains, and the TGES motif of the A domain was in place to dephosphorylate Asp307, (2) the salt bridges were broken due to rearrangements of the P domain, and the subunits KdpA and KdpB were well separated, (3) most likely AlF₄⁻ was coordinated by a Mg²⁺ instead of AMPPCP, of which no density was seen, (4) the inter-subunit channel ended at the subunit interface between KdpA and KdpB which disabled any K⁺ transfer from KdpA to the binding site in KdpB, (5) TM2 and TM4 of KdpB were moved apart by 5 and 15 degrees, and opened a tunnel reaching from the K⁺ binding site to the cytoplasm.

Based on these findings a Post-Albers type transport cycle was proposed (Figure 6) with the anomaly that the entrance for ions into the pump is reversed to the other P-type ATPases, i.e. open to the extracellular (= luminal) side in the E1 conformation, and open to

the cytoplasm in the E2P conformation. (This suggestion was first introduced by Haupt et al. (2004) J.Mol.Biol. 342:1547.)

The cryo-EM experiments and the control EPR studies were performed accurately and data analysis was conducted with state of the art techniques. The work was presented in a clear manner and is well comprehensible. Data presentation and visualization are fine, and the substantial supporting material is very helpful and appreciated.

There are, however, two major issues that need detailed consideration.

(1) Both states of the KdpFABC complex selected for detailed analysis were introduced as "a late nucleotide-bound E1 conformation" (state 1) and "a late E2-P state after ion release" (state 2). The limitation on only two states is quite arbitrary. They appeared simultaneously in the same buffer (K^+ , Mg^{2+} , AMPPNP, AlF_4^-) and an equilibrium distribution between states may be assumed, before they were shock-frosted. According to the underlying Post-Albers scheme it is obvious that both states cannot be adjacent in the transport cycle. A state without AMPPNP and AlF_4^- as intermediate is the minimum requirement. The "invisibility" of such a state could be possibly explained by the high concentration of the so-called inhibitors which would deplete the interjacent plain E1 or E2 state. However, the statement on p.4 with respect to state 1, "we can find an additional weak density in the cryo-EM map that we assigned to an AMPPCP molecule," raises an important question. The authors decided (quite arbitrarily) to pool all particle images with an appropriate arrangement of the N, P, and A domains as a single state. Is it possible that this choice produced a questionable average structure? When a distinction between particles with and without a density "assigned to an AMPPCP molecule" would be made, two different states may be revealed. In addition, a clarifying explanation to the "only partially K^+ -loaded late E1 state" may be found therefrom.

We thank the reviewer for the general positive assessment of our work. With regards to the first major point, we don't believe the limitation to two states to be arbitrary. While we cannot exclude that there might be more states in the sample, these two states are clearly the most populated ones, which were identified through extensive 3D classification rounds. We would like to point out the careful work we did during image processing. While the X-ray structure in an E1 state was initially used as a reference for 3D classification, we quickly noticed that we have different states present. In contrast to the small changes observed in the TMD, the cytosolic domain of KdpB undergoes large conformational changes. We were thus careful in not being biased by the use of one reference, which represent only one state, and therefore might favor one conformation over the other. For this purpose we made sure to undergo the whole image processing workflow with several different references, obtained during 3D-classification/refinement and which represented different states of KdpFABC. This way we could make sure to not exclude any particles or misinterpret them. This process allowed the identification of only two major populations, which in the final step were separated by a multi-reference refinement. However, if in the particle set that originated as state 1 are indeed particles with and without AMPPCP, it is at this stage impossible to distinguish them by 3D classification: First, assuming that the domains adopt the same conformation and the only difference would be the presence or absence of AMPPCP, the size of the molecule relative to the rest of the complex would be too small to allow such a distinction. Second, the resolution of the cytosolic domain is only around 4-6 Å, which we acknowledge might be an indication of flexibility. However, we would like to stress that for the assignment the EPR experiments are extremely important. They show that the E1 conformation of state 1 can only be stabilized in the presence of a nucleotide analogue, like AMPPCP, strengthening our argument. We also agree that the high concentrations of ligands/inhibitors might favor the two found states and diminish the chance of sampling an apo state of the protein. But as we now also state in the manuscript (lines 84ff) our initial aim was to stabilize an E2 conformation. A sampling of the additional E1 state was not anticipated.

(2) Concerning the proposed Post-Albers scheme (Figure 6), it has to be stated that it contravenes the functional studies from Siebers and Altendorf (J.Biol.Chem., 1989, 264:5831), in which the KdpFABC complex was maximally phosphorylated by [γ - ^{32}P]ATP in the absence of K^+ , and addition of K^+ induced dephosphorylation. They explained their extensive and consistent set of data with the "standard" Post-Albers scheme, i.e. K^+ induced dephosphorylation but phosphorylation by ATP without K^+ . Until today no functional study has been published that challenged their findings (and interpretation), although the few papers with a different mechanistic assumption just ignored these findings. The weight of structural insights from "inhibited" states of the KdpFABC complex that "resemble" functional states, as presented here, should not be deemed high enough to brush results under the carpet that represent pump function under (almost) physiological conditions. In addition, according to the proposed new transport mechanism, a "late" E1 state that precedes directly enzyme phosphorylation would require an occupation of the canonical site by a K^+ . A critical discussion of the current discrepancy of functional and the presented structural findings is the least the authors should provide. Maybe, the statement on p. 7 "the presented structures finally allow to propose a conclusive mechanism for active transport of K^+ via KdpFABC" is still somehow premature.

You are of course totally right. In our assumption we were mostly driven by our structures but we did not intend to ignore the plethora of biochemical data that has been published previously. We now state in the manuscript that there are several papers that opt for both of the possible proposed transport cycles and that further investigations are required to understand the transport cycle (lines 260ff). Particularly the data of Siebers and Altendorf seem to contradict our hypothesis but as we have learned from the now three available structures of KdpFABC this protein complex can be phosphorylated at two distinct locations. Both Asp307 and Ser162 of KdpB could take part in the amount of ^{32}P -labeled KdpFABC. For the future we therefore aim at performing similar measurements to those of Siebers and Altendorf, which we would love to correlate to transport assays to better resolve the transport cycle.

The allocation of the two states to E1 and E2 conformations has been deduced more carefully in the revised manuscript and the "late, nucleotide-bound" E1 state has been reassigned as plain E1 state. We think the empty canonical binding site in KdpB might be explainable by the proposed mechanism in which potassium ions are pushed through the horizontal tunnel by a knock-on mechanism. We think that the "chain" of ions has not reached the canonical binding site yet. However, as there is not even an agreement on how potassium ions are translocated through a selectivity filter, further studies, particularly MD simulations, are required to understand the transportation mechanism through the tunnel. The sentence "the presented structures finally allow to propose a conclusive mechanism for active transport of K^+ via KdpFABC" has been rewritten to a less ultimate formulation.

Reviewer #2 (Remarks to the Author):

The manuscript entitled "Cryo-EM structures of KdpFABC reveal K⁺ transport via two inter-subunit half-channels" describes two structures of the membrane protein complex KdpFABC in two different conformations, leading to the proposal of a novel mechanism for K⁺ transport.

The work presented appears to have been well conducted. The study is important as it reveals molecular features of the mechanism of a K⁺ pump that is unique to bacterial systems and that plays an important role in the physiology of many bacteria. Strikingly, the architecture of the complex and associated mechanism are also unusual since they display features found both in K⁺ channels and P-type ATPase pumps. The study will be of great interest to structural biologists and microbiologists in general and to researchers working on the mechanism of ion transport in particular.

I have a few problems with the manuscript.

1- The assignment of a catalytic states according to the Post-Albers cycle (state 1 according to the authors corresponds to the E1 conformation and state 2 to E2-P conformation) is an important aspect of the analysis of the structures and it should be better supported by a Figure showing a more careful comparison with existing structures of other P-type ATPases. This new figure will also help the reader that is less familiar with the intricacies of the ATPase ion pumps. In this context, the authors refer as an important aspect of their assignment the position of the TGES motif in the A domain but this motif is not clearly indicated in the figures.

First of all we would like to thank the reviewer for the in general positive feedback as well as the constructive suggestions. We agree with the reviewer and have made several changes in the manuscript. The assignment of state 1 and 2 to an E1 and E2 state is mainly based on the cryo-EM data. To highlight this we now show the positions of the N A and P domains better in Figure 1 and Supplementary Figure 5 and rewrote the manuscript accordingly. In supported of the conclusions from the cryo-EM data we compared our structures with known SERCA structures and provide superpositions and RMSD calculations (Supplementary Figure 6 and Supplementary Table 2).

To better introduce P-type ATPases in general we also included a paragraph in the introduction (lines 39ff.)

2- The assignment of the catalytic state also involves as discussion of the possible presence of nucleotide. The authors use as arguments to support this presence the existence of weak density associated to the N domain and the comparison of one of the KdpFABC structures with a structure of SERCA in supplemental Figure 5.

We have to admit that we didn't develop the assignment well enough in the initial version of the manuscript. The assignment of which ligand is bound is actually based on the interpretation of the EPR data, which we now explain in greater detail (lines 119ff). Since the cryo-EM density is very weak in this region as expected for a local resolution about 5-6 Å, we have excluded the weak densities for nucleotides and the AlF₄⁻ from the manuscript. However, to provide evidence that in general the nucleotides could be bound within the structures we docked the ligands into our models based on a comparison with SERCA structures 1T5S (E1, AMPPCP-bound) and 3B9R (E2, AlF₄⁻ and AMPPCP-bound) (see Supplementary Figure 8 and Supplementary Table 2).

The weak density for the nucleotide is very difficult to evaluate as presented in Supp. Figure 4d. I suggest the generation of a stereo-image to better present this data. The same applies to the density for the AlF₄⁻ molecule in panel e) of the same figure. See above comment.

The superposition of the nucleotide binding sites of KdpFABC and SERCA in Supplemental Figure 5c) is supposed to support the idea that they are similar. However, the superposition clearly shows that the two conformations are different and the sites do not

match so it is unclear how the conclusion is reached. See further comments on these figures below.

Part of this comment was already addressed above. However, we of course agree that none of the superpositions with SERCA structures result in a perfect match but rather demonstrate a good agreement of the overall arrangement (cf. Supplementary Figure 6). From our comparisons particularly the E1 state seems to be different from all other E1 states solved so far and also the coordination of the AMPPCP in state 2 is likely less pronounced than in SERCA structure 3B9R. However, we are confident that our EPR measurements clearly demonstrate that the assigned ligands have to be present to stabilize the achieved conformations.

3- The reasoning behind the analysis of the EPR results is a bit confusing since the authors state "However, it [AMPPCP] stabilized an E1 conformation with a narrow distance distribution between the spin-labeled residues centered at 4nm resembling the observed distances in state 1...". As far as I know a P-type pump can bind ATP (or AMPPCP) both in E1 and E2 states. It is therefore not obvious the reason why the authors have assigned the state E1 to the conformation in solution by EPR.

The assignment of state 1 and 2 to an E1 and E2 conformation is not based on the added inhibitor but solely on the relative position of the N, P and A domains, which is despite the lower resolution unambiguously resolved in the cryo-EM maps. As known for all P-type ATPases an E1 state is present when the TGES loop (A-domain) is distant from the phosphorylation site (P-domain), while it is in close proximity in an E2 state. To show this better, we have as recommended by the reviewer changed Figure 1. Additionally, we have included an extra Figure (supplementary figure 5 and 6) and RMSD data (supplementary Table 2), which compare both KdpFBAC states with 12 SERCA structures. Hence, the ligand/inhibitor we expect to stabilize which conformation is based on the interpretation of the EPR and ATPase data. As mentioned above, these experiments are now described and explained in more detail throughout the manuscript.

4- The authors should explain in more detail the rationale for assigning a K⁺ to the densities present in the tunnels observed between KdpA and KdpB? Could these densities be well ordered water molecules that are detectable even at this low resolution?

We have addressed this question in great detail but would ask you to read the extensive comments for reviewer 5 about this issue.

5- In page 6 the authors speculate that water molecules oriented by a second layer of the selectivity filter (Asn114, Gly232 and Gly345) also coordinate the K⁺. What is the basis for this speculation? Since this statement does not seem to be based on any data provided by the structure it seems that this level of speculation is unnecessary.

Since the selectivity filter of KdpFABC is very asymmetric and the coordination of K⁺ ions in the possibly up to four positions of the filter has not yet been fully described, it could be possible that the coordination pattern is not as strict as in other potassium channels. In the classical TVGYG selectivity filter each K⁺ ion is coordinated by two layers of the selectivity filter. Since the four filter loops of KdpFABC are ₁₁₂NTNWR, ₂₃₀TNGGG, ₃₄₃SCGAV and ₄₆₈NNGSA a different way of coordination is to be expected. But since we do not see coordinating waters the sentence has been removed.

6- In page 6, the authors state " suggesting the possibility of a coordinated movement of partially hydrated K⁺ ...". I am not sure what "coordinated movement" means.

With the term of "coordinated movement" we tried to explain that a knock-on mechanism is the most probable propagation strategy for K⁺ ions through the inter-subunit tunnel. This is clarified in the revised manuscript. However, we also state the particularly MD simulation will be helpful to better understand the mechanism.

A major problem I have found with the manuscript is the organization and quality of the figures that do not facilitate the analysis and appreciation of the work. In particular, the authors have opted to use a weak and strong shade of the same color in the two different structures. This choice instead of helping the visualization of the different conformations

makes it harder.

Figure 1: panel c) is very difficult to understand since I cannot distinguish the different domains from the two different structures. In addition, the authors refer to Figure 1c to indicate the differences in the relative position of the spin labels but the corresponding spheres are barely visible. I suggest that the authors use the same color scheme for the two states and simply show the two states side-by-side. Additionally the authors have used superpositions in several panels and in general these are very confusing, for example in Figure 1 panels a) and b) just show a mash of densities and helices which have little use. I strongly suggest that the authors avoid the use of superpositions (except for a few cases and in these more care should be taken to make the figure useful) and just place the two states side-side and in the same orientation.

We particularly thank the reviewer for these suggestions because they really helped us to improve the quality of our figures. For all figures the color code has been changed. In cases of superpositions state 1 is now in grey while state 2 is color-coded. Any unnecessary superpositions have been replaced.

Figure 2e: (and in other representations of state2, see below) the actuator domain (light yellow) is barely visible.

See above comment.

Figure 4: indicate TM2 and TM4 in panels a) and b) so that they are more easily related to panels c) and d) and better support the point made in the main-text.

TMs have been labeled wherever relevant.

Supplemental figure 3. In panels 3b, 3c, 3f and 3i the actuator domains are barely visible. Since the structures are presented in different panels better use the same exact color scheme across all structures to better compare positions of domains. Superposition in panel c) is not useful since it is just a mash of helices.

See above comment.

The authors also show representations that are related to each other in different figures, making it harder to understand the points raised in the text. In particular, panels that present phosphorylation at S162, the binding site of nucleotide and the AIF4- binding site are spread between Supplemental Figures 4 and 5. It would be much more reader-friendly if these panels were put together in a single figure, leaving the density for the secondary structure elements shown in Supplemental Figures 4a and 4b as a separate figure, ideally with a larger size for better analysis of density quality.

Organization of Supplementary Figures has been changed accordingly.

I also found that the main text presents a few awkward sentences that I strongly suggest should be changed:

page 2, 1st paragraph in main text

"... the highly K⁺-affine, actively pumping P-type ATPase...". Needs to be rewritten.

"...the KdpFABC complex is the only known chimera composed of 4 different subunits".

Besides the unconventional use of the term "chimera", the sentence appears to state that KdpFABC is the only known "special" complex composed by 4 different subunits. I am sure that this is not what the authors intend to say and it should therefore be changed.

Page 5, 2nd line:

" ... the phosphorylation does not lead to the initially proposed auto-inhibited...".

I suggest something more like "... phosphorylation at S162 in different conformational states of KdpFABC does not fit with the previously proposed model where phosphorylated S162 was thought to lock the A and N domains in an auto-inhibited conformation."

Page 7, 1st paragraph:

"...composed of the two half channel is promoted by the absence of any of the previously suggested conformational changes within KdpA and KdpC."
Probably say that "...composed of two half channels does not involve the previously suggested conformational changes...".

Page 7 2nd paragraph:

"After decades of studies, the presented structures finally allow to propose a conclusive mechanism for transport of K+...".

This rather grandiose sentence would greatly improve if the word "conclusive" was omitted. Whether this important study is the defining work on the mechanism of KdpFABC transport mechanism will be determined in the future with other studies.

Page 7 2nd paragraph:

"Other than expected, not solely the channel-like subunit B facilitates K+ translocation, but rather the combination of two joined half-channels ...". This sentence could be simplified.

All suggestions were incorporated.

Reviewer #3 (Remarks to the Author):

KdpFABC is an important K^+ import system helping bacterial cells deal with several environmental stressors. A crystal structure of the entire complex (actually a mutation, R116, with decreased activity) has been determined and published last year (Huang et al., Nature 2017).

A sentence mentioning the mutation and the accompanying decreased affinity for K^+ has been included.

In the present manuscript, the authors present two cryo-EM structures of the KdpFABC complex, which they link to two distinct states of the transport cycle. The cryo-EM analysis appears straightforward and robust and the work is supported by ATPase assays along with EPR and MS analyses. The novelty of the current results rests on the proposal of a new mechanism for cytosolic K^+ release, suggested previously (Huang *et al.*, 2017) to occur through the channel (KdpA) pore. In the present work the authors suggest that the release instead occurs through a newly observed tunnel in KdpB leading from the canonical cation binding site in KdpA to the cytoplasm and forming upon transition to the proposed E2P state. The Stokes group (Huang et al, 2017) had previously proposed that occlusion of substrate binding sites occurs through a rearrangement of KdpC at the periplasmic side and by conformational changes of the so-called coupling helix/loop of KdpA at the cytoplasmic side. In contrast, here the authors promote the idea of occlusion of the K^+ binding sites in KdpA from K^+ ions in KdpB by a closing of the previously observed horizontal tunnel connecting the two interacting proteins. The authors further propose that unlike prototypical K^+ channels, the KdpA channel remains closed as no conformational changes of the proposed gating elements are observed, and that K^+ instead is shuttled through the open inter-subunit tunnel to the canonical substrate-binding site in KdpB by a 'knock-on' mechanism. If true, this would represent an exciting, albeit unusual finding. The validity and general acceptance of this proposal would be greatly aided by a transport assay some evidence directly supportive (in particular showing transport of K^+ without an opening of the cytosolic gate of KdpA). Without such a demonstration, the conclusions remain very speculative. Unfortunately, the presentation of the paper is at times confusing and discrepancies between the current structures and previously determined crystal structures need to be addressed. The description of some of the methodology is also confusing.

We apologize that the initial version of our manuscript has caused some confusion and hope to have solved this issue with the revised version by better dissecting our conclusions step-by-step. Furthermore, we now better support our conclusions with published biochemical data; for example we highlight several mutations that previously were shown to affect ion selectivity, ATPase activity or potassium ion transport, which all align with the here suggested transport pathway.

Specific points:

1. The proposed mechanism stipulates that the channel (KdpA) remains closed throughout the transport cycle. There needs to be some experimental demonstration of this to support such a conclusion. The authors should consider (disulfide) cross-linking to immobilize pore linking helices and/or the gating helix to prevent channel opening and to confirm that transport activity is maintained when channel opening is prevented.

We agree that future experiments need to be conducted to prove the hypothesized new transport mechanism. However, this is in our opinion beyond the scope of this manuscript. Nonetheless, the fact that no movements of KdpA and KdpC are observed in the now three distinct available structures of KdpFABC, is very indicative that their role in the transport mechanism must be different than previously assumed.

2. The physiological relevance of the simultaneous use of non-hydrolysable ATP analog and the Pi mimic is questionable. Why was ADP excluded and what happens (functionally) in the presence of ADP/ALF4-?

As we now state in the paper we aimed at solving the structure of KdpFABC in an E2 state since the previous crystallographic structure was stabilized in an E1 state. For this reason we choose a SERCA structure [3B9R] as model, which was stabilized in a similar way. ADP was not excluded but we choose to try something else. In ATPase assays, performed as the ones described in the manuscript, the mixture of ADP and AlF_4^- reduces the activity to 10-20% of wt activity and could in future experiments as well be used to stabilize KdpFABC in another conformational state.

3. The continuous tunnel (along the selectivity filter and horizontal tunnel between KdpA and kdpB) was observed in the previous crystal structure (after exclusion of the mutant R116 sidechain, which mimics a K^+ ion at S1). The assignment of K^+ ions to any density in the current work is speculative and figures depicting bound K^+ ions (e.g. Figs 2b-d, g), made to look like Fo-Fc difference maps, could be misleading in the absence of expanded figures showing density for all surrounding ligands (basically, it could be noisy density made to look like specifically bound elements based on the density threshold and carve around the placed ion). The same is true for proposed nucleotide / AlF_4^- density, although the authors explicitly state that the density is weak. The EM density evidence is therefore currently insufficiently demonstrated.

About the assignment of potassium ions, please see extensive comment for reviewer 5. A Supplementary Figure showing the densities for the ions as well as their surrounding has been added, to prove that they are well defined and do not represent noise. In the case of the nucleotides and AlF_4^- , we have excluded the discussion about those weak densities from the manuscript since the local resolution is only about 5-6 Å and in fact very difficult to interpret. Main evidence for the inhibitors bound actually derived from EPR experiments, which we now explain in much more detail (see lines 119ff.). Additionally, to show that in principle spaces and coordination for the molecules is given in the structures we docked inhibitors into state 1 and state 2 guided by available SERCA structures (cf. Supplementary figure 8).

4. The authors should consider focused refinement of subdomains/domain pairs in an effort to improve the resolution to clarify binding of nucleotides/ AlF_4^- . As it stands, the placement of nucleotides/ AlF_4^- is highly speculative with what appears to be very noisy density in the relevant regions.

In fact we have tried it but unfortunately it did not help in our first trials. We assume that the cytosolic domain is too small, at the edge of the extracted box and too less resolved at this stage. We aim to follow up on this strategy by using upcoming new available software, and tackle this type of challenging data. Therefore, we no longer show the densities of nucleotides and AlF_4^- in the revised manuscript. (see above comment).

5. The crystal structure of the complex excluded K^+ binding at the canonical site, which was instead proposed to be occupied by a water molecule. The currently proposed mechanism necessitates K^+ binding at this site, which would lead to introduction of a positive charge next to K586. The authors should discuss this discrepancy in greater detail (as mentioned in point 2, assignment of density within the tunnel as K^+ ions is speculative).

The superposition of canonical ion binding site of KdpB in state 1 and state 2 depicted in a new Figure 4c, shows that the side chain Lys586 is oriented away from the binding site in state 1, where we would expect potassium binding. Only in state 2 the side chain of Lys586 protrudes into the binding site, possibly pushing a potassium ion into the cytoplasmic exit tunnel. We thank the reviewer for pointing this out and we do now discuss the role of Lys586 and Asp583 in more detail throughout the manuscript.

6. In the current scheme, it is unclear how K^+ would be released against a concentration gradient from the E2-P state considering knock-on (invoked here as the primary mechanism driving K^+ binding and transport) is prevented by the occlusion of the canonical site from ions bound in KdpA. The authors should explicitly present a superposition of the relevant KdpB residues in both states.

As we now state more explicitly in the text, we consider a knock-on mechanism for K^+ in the outward-facing half-tunnel. However, this is of course speculative and needs to be addressed probably by MD simulations in the future. The release of K^+ to the cytoplasm in an E2-P state would be a consequence of the rearrangements during the E1P/E2P transition that disrupt the high affinity binding site (see above comment). This mechanism would in fact be comparable to other P-type ATPases. On another note, we would like to mention that the previous model which assumed an opening of a channel makes it even harder to explain active transport of K^+ against such a big gradient. The model proposed here brings back a "classical" primary-active ion pump mechanism.

7. The entire purification of EM samples is done in the absence of K^+ , which is introduced in combination with the inhibitors prior to grid preparation at a lower concentration than the Na^+ present in the purification buffer. In the absence of K^+ , many K^+ channels have been known to conduct Na^+ . Can the authors rule out Na^+ specific effects based on ATPase stimulation (or lack thereof) for the wild type protein used here at different K^+/Na^+ ratios? It has been shown several times that Na^+ is not able to stimulate ATPase activity of KdpFABC and that it is not transported. In fact, this is the case for all group I elements. This point is now also discussed and referenced in more detail in the revised discussion of the manuscript.

8. The authors conclude that KdpC likely plays only a stabilizing role. How likely is this considering the extensive hydrogen bonding contacts between KdpA and KdpC, which would likely couple any changes in the channel domain to conformational changes in KdpC as previously proposed based on the crystal structure (Huang et al, 2017)? This should be discussed further in terms of known KdpC mutations that alter transport and ATPase activity.

As we have postulated in the manuscript, KdpC could not only have a stabilizing role but likely contributes to increase the K^+ affinity of the complex. As seen for β -subunits of other K-transporting P-type ATPases that are structurally similar to KdpC. Most of the mutations and truncations of KdpC have been performed under the hypothesis of it being a catalytic chaperone during ATP binding in KdpB and experiments mostly involved only soluble parts of KdpC and KdpB. Therefore with today's knowledge we find it very difficult to draw conclusions from these experiments, because predominantly ATP binding of KdpC and not hydrolysis or substrate transport of the whole complex had been assumed and examined.

Reviewer #4 (Remarks to the Author):

Stock and colleagues present two cryo-EM structures of the prokaryotic P-type ATPase, KdpFABC determined in the presence of AMPPCP and AlF_4^- at resolutions of 3.7 and 4.0 Å. Comparison of these structures with an existing crystal structure of KdpFABC reveals that these structures represent novel states for the transporter. Comparisons with structures of other P-type ATPases suggest that state 1 corresponds to an E1 conformation while state 2 corresponds to an E2-P conformation. ATPase assays and EPR distance distributions, which demonstrate that AMPPCP and AlF_4^- inhibit ATPase activity and stabilize the transporter in E1 and E2-P states, provide additional evidence for the assignments. In the cryo-EM density map, the authors note the presence of densities that they have assigned as ordered K^+ ions. The presence of these density peaks along with measurements of a previously described cavity extending between KdpA and KdpB allow the authors to delineate a putative ion translocation pathway that extends from the selectivity filter region of KdpA through KdpB into the cytoplasm. Based on the novel ion conduction pathway and on conformational changes in the P domain of KdpB subunit, the authors present a transport mechanism where K^+ ions are concentrated by a high affinity site in the KdpA selectivity filter, moved through the pathway towards KdpB and then actively transported through KdpB *via* an alternative access mechanism.

The authors conclusions are well supported by multiple lines of evidence. However, the presentation of the data makes it somewhat difficult to follow the authors' argument for their transport mechanism.

Several changes to the manuscript could be made to improve the presentation of the data.

1. The authors title Figure 1 as an "Overview of KdpFABC cryo-EM structures in different conformations". However, the figure is lacking a panel presenting an overview of the structure of KdpFABC to provide readers not accustomed to analyzing P-type ATPase structures a frame of reference for understanding the different conformations displayed by the two structures and for the detailed analyses presented in the subsequent figures. Figure 1 was changed and the cylindrical representation in panel a and b give a good overview, with panels c and d showing the clear distinction of an E1 and E2 state.

2. Continuing from point 1, maintaining a single color scheme throughout the manuscript would aid readers in understanding how movements of the transporter during explain the transport mechanism. The color code for states 1 and 2 is now identical throughout the manuscript, only in cases of superpositions state 1 is shown in grey while state 2 is color-coded.

3. In Figure 1c, it is nearly impossible to identify the spheres used for the spin labels in the image. Please use an alternative color or highlight the spheres in another manner. New Figure 2 a & b highlight the positions labeled for EPR.

4. There are several statements in the manuscript that are not clear. For example, it is written on page 5 "Hence, the EPR data demonstrate that AMPPCP is required to achieve state 2, although it is not visible in the cryo-EM map. Notably, also identical structures of SERCA in an E2-P conformation with [3B9R] and without [3N5K] AMPPCP bound were solved". We have rewritten larger portions of the manuscript to make it easier for readers to follow our line of argumentation concerning assignment of states as well as placing of ligands/inhibitors.

Comments:

1. The authors should more fully address the mechanisms by which nucleotide state determines that state of the transporter and how changes in nucleotide state drive the reaction. Also, why is AMPPCP required for stabilization of the transporter in state 2, yet it

is not present in the structure?

The allocation of state 1 and 2 to an E1 and E2 state has been entirely based on the relative orientation of the N, P and A domains revealed in the cryo-EM maps. The assumption of which inhibitor is bound, was based on the EPR data. These results were further supported by extensive comparisons with SERCA structures and RMSD calculations. We thank the reviewer for pointing this out. We have included additional data and discuss this issue in more detail throughout the manuscript.

2. In Figure 2, the authors present density peaks that they have attributed to K^+ ions. While the peak in panel b appears to be partially coordinated by Q116, N239 and G468, the peaks in the remaining panels are poorly coordinated or not coordinated at all. At the intermediate resolutions of these reconstructions, how do the authors justify that these density peaks correspond to K^+ ions?

Please see extensive comment for reviewer 5.

3. The authors suggest that the KdpC subunit serves to increase the K^+ affinity of the transporter in a mechanism similar to the β subunits of the Na^+ , K^+ ATPase. Such an argument would be strengthened by an analysis of the electrostatic surface potential of the opening of the transporter or by functional assays demonstrating that mutation of specific residues in KdpC inhibits K^+ uptake against large K^+ concentration gradients.

Indeed, we have generated an electrostatic surface map of KdpC, but unfortunately it was not conclusive. Testing KdpC mutants that alter K^+ affinity is required to strengthen this hypothesis, but this is beyond the scope of this work. We hope the reviewer acknowledges that we do not claim to have any prove of it but that it is a mere and valid suggestion in the final discussion, based on the location, immobility of the domain at different states, and its resemblance to other p-type ATPases.

Reviewer #5 (Remarks to the Author):

First of all I would like to congratulate the authors on a beautiful piece of research revealing two novel conformational states ($\sim 3.7 \text{ \AA}$ and $\sim 4.0 \text{ \AA}$) of the potassium transporting Kdp complex using cryo-EM. Both states are key intermediates of an active transport cycle that has been the focus of much speculation previously in the literature and which we get a glimpse at now for the first time.

The Kdp system is an highly interesting K-transport system since it combines two very distinct super-families of transport proteins with completely disparate modes of action. Namely, the SKT channels and the P-type ATPases. The fact that Kdp only function when both of these distinct components are together present a unique and fascinating case study in evolution, and much can be learned about both classes of proteins from this obligate complex and its mechanistic function. This is well exemplified in the presented manuscript that bring to life completely new ideas and suggestions on how both channels and pumps might function mechanistically, that would not have been thought of if one focused solely on one type of transport.

There can be no doubt in my mind that the work and the topic are highly relevant and appropriate for the Journal, and overall, this reviewer strongly recommends publication in Nature Communications.

We would like to thank the reviewer for his enthusiastic support of our data. Furthermore, we are grateful for his versatile and very helpful comments. We very much appreciate the effort he put in improving the presentation of our data!

However, there are multiple things that I would like to see addressed before publication. Some are related to the method and must be addressed prior to publication, and some are related to the analysis and conclusions drawn from the analysis and should preferably be addressed, or at least better explained in the manuscript. I do not think that any follow up experiments are needed for now.

As a short insert, line numbers would have been nice, and allowed more detailed comments on actual wording.

Line numbers were inserted.

A few textual/figure notes:

Page 2 bottom (line 5 of text): KdpFABC is not a P-type ATPase, KdpB is. Please make sure to refer to KdpFABC as a complex, not a P-type ATPase. This language mix-up is common in the Kdp literature, but should be avoided for clarity. In a similar vein KdpFABC would also not be called an SKT protein.

Page 2: The SKT super-family name should be introduced with KrtB and TrkH in the beginning of the manuscript to avoid confusion, when KdpFABC is introduced, since KdpA is also member of the same group of proteins with a similar physiological function (potassium homeostasis).

Page 3 top: When the gating loop is introduced it should be stated that it is found in the D3 repeat.

All above raised points were addressed.

Page 8 middle: "... as proposed 30 years ago.." Yes, but for Na/K ATPase, and H/K ATPase, not for KdpC. Please clarify this sentence. Btw, why is it relevant to know that this was proposed 30 years ago?

In fact, the authors of the mentioned paper explicitly mentioned KdpFABC! We still are fascinated by this and surprised that the similarity to \$\beta\$ -subunits had been ignored in the literature till then. Isn't it surprising that despite several lines of evidence for the localization of KdpC (also the positive inside rule by Gunnar von Heijne applies) it has been assumed wrong until the recently published first structure of KdpFABC by the Stokes lab? We think we clarified this point in the revised manuscript.

There are no clear figures on the domain movements of the ATPase between the states. Only an overlay in fig. 1 and an overview in supp. fig 3. This should be amended. It is very hard currently to visualize the P domain movement and the position of the TGES loop. Figures are generally very confusing. They need more labels, a bare minimum would be to label the TM helices (all figures, but grievous examples would be fig. 2b-d, f-g and fig. 4). Fig 1 in particular is very confusing. The authors should separate the states to make it easier to see them, and remember that many people are colorblind. The two colors for the states are very hard to distinguish (I showed that figure and Supp fig 6 to a color blind colleague, and he could not see that there were two states, not identify the red line in the panels of Supp fig 6).

There are 3 main figures showing the tunnels (Fig 2-4). This is overly redundant. I suggest the authors merge figure 3+4, and move parts of them to supplementary.

The point of figure 5 and that one sentence on page 8, that KdpC could function like the beta subunit of the Na/K ATPase, is extremely speculative and irrelevant for the rest of the paper. It should be either deleted or moved to supplement.

In fig 6 you should highlight the two states you have models for. In Fig 2 you should explain what the red line is for (1.4 Å for water or for K ion radius?).

Supp. Fig 4 is the best figure of the manuscript in my opinion.

All points raised above were addressed with the new figures.

On the methods:

Cryo-EM: Generally the cryo-EM work is well described and everything seems to have been done correctly as far as I can tell from the description. However a few things need to be clarified. In Supp. Fig 1d, the flowchart need to explain where the initial reference state 1 and state 2 came from for the first 3D classification. It should be noted how many ab initio models were used or if at all. It is unclear when the Kdp crystal structure was used in the process. No mention is made of a low pass filtering of the input model, but I assume that this was done?

Indeed, the X-ray structure [5MRW] was used as initial model for the first round of 3D classification. Throughout the image processing workflow the best class of a 3D classification or refinement job was used as the subsequent reference (at a later stage

even with two different references representing the two different states of the protein). Using this iterative process, the first reference used only needs to have the rough dimensions of the target protein. Thus, it is less informative to include it in the workflow. However, it is well described in the Material and Methods section (Cryo-EM image processing) and in supplementary table 1. The references were always low-pass filter to 40-50Å.

Why did the authors use two different B-factors for sharpening and for the generation of images. E.g. State 1 uses -154 Å² but -205 Å² for the figures and manual inspection. The rationale should be explained.

The stated sharpening factors are applied globally to the entire map. However, since the map shows some heterogeneous local resolution it can be beneficial to calculate maps at different b-factors. As such higher resolved regions benefit from a stronger sharpening, whereas less resolved regions benefit from blurring. This approach can be used during manual model building or generation of images. As the transmembrane region is better resolved, some of the images were rendered with a higher sharpening. It should be noted that the actual model refinements were all carried out with the lower b-factor and that additional sharpening did in any way create a too noisy map. Both maps will be deposited.

Supp fig 2c: the A and N domain appear to be missing? What happened to the contour level here?

Indeed, for state 1 the A and N domain are less resolved and became less visible at the used contour level. This is a consequence of the heterogeneous local resolution, where the cytosolic domain is less resolved than the transmembrane region. We have included an additional panel in supplementary figure 2, with the cytosolic domain at a different contour level allowing a better inspection of the region. Please note that this contour level cannot be used for the entire protein as the density for the detergent micelle would obscure the TMD.

The authors show the sample to be phosphorylated at S162 in the two observed cryo-EM states. This is important for the downstream analysis. In the crystal structure paper (Huang *et al.*, Nature, 2017) it is demonstrated that the phosphorylated sample has no ATPase activity, and the observed activity is therefore proposed to arise from a fraction of the purified sample that is not phosphorylated at S162.

In the manuscript the MS data (supp table 2) seems to show the same pattern. Namely that a fraction of the purified sample is not phosphorylated. I assume that this fraction is what gives the observed ATPase activity in supp table 3. If so, this should be explained, or alternatively, if the authors truly believe that the Kdp complex can exhibit ATPase activity while being in the S162 phosphorylated form this should explicitly be stated, and then backed up by some evidence. That scenario would be hard to imagine, given what we know from SERCA and other P-type ATPases about the function of the TGES loop, and does not fit the data from the Huang *et al.* paper.

The paper of Huang *et al.* is very conclusive concerning the Ser162 phosphorylation. It is clear that the amount of protein phosphorylation correlates to the proteins activity. Furthermore, the authors state that KdpFABC is stabilized in the crystallized E1 conformation due to salt-bridges of Ser162-P with Lys357 and Arg363. The point that we wanted to make is solely that KdpFABC in state 1 and state 2 does not exhibit the same interaction between Ser162-P and Lys357 and Arg363 and hence is stabilized in a different way. Consequently, the previous assumption that the phosphorylation of Ser162 arrest the protein in an auto-inhibited locked conformation does not hold. Since both states are phosphorylated the protein might be able to undergo E1 to E2 transitions. These points are now described more accurately in the manuscript (lines 111ff.).

I tentatively conclude that the states observed in the EM maps are phosphorylated and inhibited states. It would be very interesting to hear the authors view on this aspect of their data, and furthermore it should be much better explained in the manuscript that the states observed are inhibited states, if the authors believe so.

In fact, we are not fully convinced whether a phosphorylation of Ser162-P fully inactivates the protein by arresting it in an inhibited state. Since KdpFABC is phosphorylated in all three known structures, the question is thus pertinent how it could still cycle between these states if it is fully inhibited. And are these conformational changes coupled to ATPase and transport activity? Further, considering the residual ATPase activity (which we acknowledge might originate from non-phosphorylated S162) and the sampled distinct states observed in EPR data in absence of inhibitors, we could equally envision that Ser162-P might significantly slow down transport activity but not fully abolish it. We simply do not know the answer but would be happy to find out in the future. We included a sentence about this open question in the revised manuscript (line 115f.)

It would also be interesting to know why the authors decided to use an inhibitor mix of AIF4 and AMPPCP for the work. I assume this was based on the 3B9R structure of SERCA? Exactly, we aimed in trapping KdpFABC in an E2 state analogously to seen for SERCA. We better describe this in the manuscript now (lines 84ff.).

EPR:

The EPR experiments are described in a somewhat confusing manner. Generally the experiments does not really contribute to the story, and should be re-delegated to supplementary material (move panel fig. 1d to supplementary).

We think that the EPR data is very important for our story and admit that they were not ideally explained in the first version of the manuscript. We now clearly state that the assignment of an E1 and E2 state derive from the relative position of the N, A and P domain shown in the cryo-EM maps. However, the assignment of which inhibitor was required to stabilize which state, and thus which exact conformations the two states represent is based on the EPR data, underlining its importance for the manuscript. We have now revised this point and made it clearer throughout the manuscript (lines 119ff.).

First a discrepancy: In Fig. 1d the E1 grayed out area is centered on 4 nm, and does not match the red curve in supp fig 6d (with 5 mM AMPPCP) which is centered on 3 nm. How is the grey area denoted E1 calculated?

The steps were as follows:

- For state 1, state 2 and the crystal structure the distance between the two labeled residues (G150C and A407C in KdpB) was calculated using a rotamer library approach implemented in the MMM software, for state 1 and 2 these are depicted in Figure 2a and b, and as grey dotted lines in Figure 2c in the original version of the manuscript
- Next, EPR traces under different conditions were recorded and compared to the calculated distances, and the tested inhibitors were correlated to the states they inhibit
- Since the states were already assigned to the conformations they stabilize by the comparison to SERCA structures, EPR was solely used to determine which inhibitors of the added mix stabilized which state/conformation

We assume that the discrepancy for state 1 between the calculated distance distribution (centered at 3.5 nm) and the EPR trace recorded with 5 mM AMPPCP (centered at 4 nm) results from MMM calculations on state 1. Already small changes in side chain orientations cause changes in the resulting distance distributions. The local resolution of state 1 of the cytoplasmic domains is only between 5 and 6 Å. This can easily cause the discrepancy between predicted and measured distance distributions as seen here. We detail this problem in the revised manuscript (lines 133ff.).

As I understand it from especially Supp fig. 6, the observed state 1 and state 2 are used to calculate the red predicted distance distributions in panel d of supp. fig. 6, but in fig. 1 the same prediction is then used to denote the gray areas as E1 and E2-P instead of state 1 and state 2. This is almost circular logic. The EPR experiment does only show that AIF4 is needed to reach state 2 in the sample, not that state 2 is the E2-P state.

We do not totally agree on your interpretation of our EPR data. The general assignment of E1 and E2 conformations was mostly based on the orientation of N, P, and A domains

towards each other and a comparison of our states with SERCA structures in different E1 and E2 states. To then actually conclude whether the added inhibitors were needed to stabilize the solved states we used EPR experiments. Consequently, the conclusion that inhibitors were needed to stabilize a distinct conformation suggests that they were actually bound allowing a further classification of our states. From the EPR experiments we could deduce that AMPPCP is sufficient to stabilize state 1. Thus we suggest that state 1 is an AMPPCP-bound E1 conformation. State 2 is only sufficiently stabilized in the presence of both AlF_4^- and AMPPCP suggesting that it represents an E2-P conformation (with AlF_4^- as phosphorylation mimic) with additional AMPPCP bound in the modulatory site. Further docking approaches guided by SERCA structures support these assumptions showing that the molecules could be coordinated within the two states in principle. We hope that this is now explained in a comprehensible manner in the revised manuscript (lines 119ff.).

Calling it E2-P in Fig. 1d (while it may be true) gives the impression that the E2-P conformational conclusion is based on more than the structural comparison of state 2 to SERCA crystal structures. Thus these labels for the grey areas in fig. 1d should be something like 'theoretical/pure state 1' 'theoretical/pure state 2' or similar.

Grey areas showing states are removed, instead we are comparing our experimental data to the predicted distance distributions using MMM. In the revised version grey areas were now implemented to mark non-trustable distances due to experimental limitations. This is detailed in the text.

One speculates where in this spectrum the Kdp crystal structure would be. A quick measurement reveals that in the crystal model the relevant distance is ~ 3 nm. Right between the distances of 4 nm for state 1 and 2 nm for state 2. The crystal structure was solved from a sample with 5 mM AMPPCP, and so if measured would presumably show a trace similar to the orange trace in fig 1. However the actual crystal structure did not contain AMPPCP in the binding site. It would be informative to see the theoretical curve for the crystal model as well in supp. fig 6d.

The MMM calculation of the crystal structure was added to the first panel of new Figure 2 and relate rather to a distance distribution around 2.5 nm. This distance is covered in the apo condition. We added a sentence about this observation in the revised manuscript (lines 129ff.).

Supp fig 6 and fig 1: Out of interest, why does the x axis start at 1.5 nm?

Pulsed EPR measurements are based on the separation of spin fractions by the given pulse sequence. It is then measured how the two spins affect one another. At distances below 1.5 nm the spins cannot be separately excited due to experimental limitations, thus 1.5 nm up to now is the minimal distance measurable by pulsed EPR.

On the analysis:

All of this bring us to THE key analysis. What are the two observed states?

The authors believe that they are observing E1 and E2-P. This is likely correct (but arguably an S162(P) inhibited E1 and E2-P). However the EPR experiments do not prove this as argued above, and the actual EM density for the inhibitors is less than convincing (Supp fig. 3d and e). I am somewhat shocked that it is apparently acceptable in the EM world to model small molecules into density of this quality, but I am glad the authors show the density so it is possible to evaluate. This should serve as inspiration to other cryo-EM papers. According to Supp Fig 2 c and d the actual resolution in the area of the maps are ~ 6 Å so the lack of definition is not surprising.

In the end, the only real argument that the observed states are E1 and E2-P is the superposition of the states with SERCA E1 and E2-P, as far as I can tell. And this will have to be done in the cytosolic domains of KdpB, which are the parts of the EM maps that are the least well defined (Supp fig 2c+d). Thus it is unacceptable that this superposition is not shown in any figure, nor is the details of how the superposition was done explained, or the RMSD value of the superposition reported.

The only superposition figure is a detail of the AMPPCP site found in supp fig 5c, and there the overlap actually looks pretty bad.

The reviewer is correct and we did not have any intention of modeling the ligand only based on the weak cryo-EM density at such low resolution. We have now tried to make the point clearer. The assignments that state 1 is an E1 state and state 2 an E2 state derived only from the cryo-EM data, which reveals unambiguously the relative position of the N, P and A domains to each other. Further, we have now included RMSD data from a comparison of both states to several SERCA structures in the manuscript and also supply a Figure with superpositions (Supplementary Figure 6 and Supplementary Table 2). The interpretation which inhibitor was required to stabilize and is thus most likely bound at which state is based on the interpretation of the EPR data and to some extent on the ATPase activity studies. Superpositions were then used to dock the ligands into their putative location at the P and N domains. Here, we now omit to show any weak density, which as the reviewer correctly states, is not sufficient neither provides the required level of detail to draw any conclusions (Supplementary Figure 8). We hope we could clarify these points much better in the revised manuscript and we would like to note that we will not include the ligands in the deposited structure.

Along the same line, it is very unclear in the manuscript which SERCA model was used for the comparison to denote conformational state. Only in the methods section are the pdb-id's 1T5S (SERCA E1-AMPPCP) and 3B9R (SERCA E2-P) mentioned, but here they are used to guide the initial model building of AMPPCP and AIF4 only? I assume (since it is not explained) that these pdb-id's are then also used later to conclude on the conformation state?

See above comment. Structures [1T5S] and [3B9R] were used as guides during docking of the ligands while several structures of SERCA were used for state determination (see Supplementary Figure 6 and Supplementary Table 2).

How many other conformational states of SERCA or other P-type ATPases did the authors try before concluding that 1T5S and 3B9R were the best fit?

In total we have compared 12 SERCA structures to KdpFABC states 1 and 2 (Supplementary Figure 6 and Supplementary Table 2). SERCA-[3B9R] showed the lowest RMSD when compared to state 2 and it was obtained at the same inhibitor conditions. While for state 1 the structure of SERCA [1T5S] did not reveal the lowest RMSD it was still used for ligand docking as it was stabilized with the same inhibitor cocktail as used in our study.

If 3B9R is the model we are comparing to for state 2, the following comes to mind. In 3B9R the E183 from the TGES loop is 2.4 Å from the coordinating water that will coordinate the AIF4 (it is a transition state analogue), and is key for the coordination. What is the distance of the equivalent residue (E161) in state 2? This is also related the phosphorylation observed on S162. Does the TGES loop actually have the same position related to the AIF4 as in SERCA E2-P transition state, or do we have an intermediate state where the AIF4 is bound but not really mimicking the E2-P transition state? How sure can the authors be, if there is ~6 Å resolution in this area? How does the model fit the density here?

We are very confident that state 2 is an E2P conformation for several reasons:

- as explained above, the relative position of the cytosolic domains unambiguously determined by the cryo-EM map reveals that it is an E2 state. The interpretation that it most likely is an E2P state is based on the EPR data and detailed comparison with known SERCA structures.
- the distances from the Glu to the central Al are 4.4Å in the case of state 2 and 4.1Å for [3B9R]
- the TGES loops are oriented in exactly the same manner
- the P domain does have better local resolution than N and A domains, around 4 to 4.5Å (see Supp. Figure 2d)

More figures and some RMSD numbers would help to understand how well state1/2 fits E1/E2-P as expected from homology to SERCA.

See above and new Supplementary Figure 6 and new Supplementary Table 2.

The whole analysis is based on the correct identification of the states so this is a key analytic step, and it should be much better explained in text and figures how the conclusion of E1 and E2-P was reached!

Once again, we would like to stress out that the main conclusion drawn in this manuscript is the fact that we have obtained an E1 and an E2 state of the complex, which allowed to identify an entry tunnel and an exit tunnel. This conclusion is unambiguously supported by the cryo-EM maps, but not dependent on the exact ligand bound nor on the potassium ions in the tunnel.

We hope that the new manuscript gives enough additional and revised figures, as well as explanations in the text to follow our line of argumentation to convince reviewer 5 of our conclusions.

In the end, I do think that E1 and E2-P are likely correct, but it took some time for me to get there due to the lack of clarity in the manuscript in this part.

In particular I need a figure of the TGES loop and its location in relation to Asp307 of the P domain in both states and the matching map density. This is very hard to see from the current figures (except supp fig 5b, a panel that should be part of the main figures).

Figure added, see Figure 1 c,d and supplementary figure 5.

The tracing of the tunnel or tunnels then become the main focus of the rest of the manuscript. Generally speaking there is just too many figures of the tunnel features. I strongly recommend the authors merge at least figure 3 and 4, and use the extra space to show of the other features described above.

We have revised all our figures and changed the order as suggested.

Figure 2 show the exciting observation of some spherical density in the tunnel. These are modeled as potassium ions and naturally are of particular interest.

It is not clear to me that a potassium ion would give a stronger peak than carbon or oxygen (eg water) in an EM potential map, in particular a map that has been blurred. This is not crystallography where the signal (sigma level) is proportional to the number of electrons.

This made me curious, and I looked into the literature on the topic, and found a very interesting paper from Wang et al. IUCrJ 5 (2018) "Identification of ions in experimental electrostatic potential maps" (<https://doi.org/10.1107/S2052252518006292>).

I wonder if the authors could use the techniques described in this paper (of blurring the maps) to help strengthen their argument that the observed peaks are indeed a positively charged potassium and not eg. water or part of a larger molecule such as lipid or detergent?

We thank the reviewer for pointing out the interesting paper that describes how one can attempt to identify ions in cryo-EM maps. Indeed, we could verify the method on a variety of maps with different cations and anions and we do confirm that in most cases it helps in distinguishing ions, as cryo-EM densities of cations tend to become stronger when the map is blurred while anions become weaker. Figure 1 shows the same procedure for the cryo-EM map of state 1. To our surprise and contrary to what we expected, the densities, which we assigned to potassium ions, became rather weaker when blurring the map. We have thus analyzed other cases and found studies of potassium channels where the structure was solved by cryo-EM at similar resolution, and which showed a similar behavior as described in our case (1: The K_{ATP} channel, EMD-7338, ref Lee et al., 2017, eLife; 2: The Slo1 Ca²⁺-activated K⁺ channel, EMD-8410, ref Tao et al., 2017, Nature; 3: The SK4/calmodulin channel, EMD-7538, ref Lee et al., 2018, Science). Figure 1 shows that, like for KdpFABC, the density assigned to potassium ions in the selectivity filter become weaker when blurring the map and stronger when the map is sharpened. It is currently not elusive to us

what the basis for this discrepancy is. It could be a characteristic of potassium ions (in particular their coordination in potassium channels) or due to the low resolution of the maps (note that the experiment described by Wang et al. is based on simulations or cryo-EM maps better than 2.5 Å).

All together, we do believe that the unassigned densities rather corresponds to potassium ions than to water molecules because: a) potassium ions have a higher atomic number ($z=19$), and will thus scatter electrons stronger than water molecules which are composed of oxygen ($z=8$) and hydrogen ($z=1$); b) while the resolution in the TMD regions is partially below 3.5 Å we do not believe that it is yet sufficient to resolve water molecules; c) one wouldn't expect to have a water molecule coordinated at the selectivity filter. Finally, there is no reason to assume that the densities might correspond to lipids or detergents as the densities are buried in the protein, too small and the tunnel too narrow for this. While we agree that at this stage there is no absolute prove that the densities originate from potassium ions, we conclude that this is the most reasonable interpretation of the results.

Figure 1: Impact of sharpening on the cryo-EM densities of potassium ions. a-d show the cryo-EM densities in state 1 that were assigned to potassium ions in the selectivity filter (green lines) and in the tunnel. The densities were additionally blurred (b, blue), plotted at with a b-factor of -154 (c, purple) or additionally sharpened (d, red). All densities were plotted at 3σ . e-h show the cryo-EM densities of the selectivity filter with three assigned potassium ions of the human K_{ATP} channel (Lee *et. al.*, 2017, eLife) solved at 3.9Å. The densities were additionally blurred (f, blue), plotted at the original b-factor (g, purple) or additionally sharpened (h, red). All densities were plotted at 18σ . e-h, For clarity only the density of two opposite chains of the selectivity filter are shown. a-h, for clarity potassium ions are displayed with a radius of 0.5Å.

I also wonder if the fig 2b-d peaks have been cropped so that the surrounding density is not visible? This is not explained in the figure text.

The densities were cropped in this image. To show that this was not noise we have included the same image including the surrounding density in an extra supplemental figure 11.

For now, the modeling of these 3 peaks as specific atom types should be done very carefully and with great respect to the surrounding chemical environment. Since this again is a key aspect of the analysis much better figures should be presented. Fig. 2 Panels b-d are not sufficient to properly evaluate the chemical environment of these peaks from the readers perspective.

Site 1 (fig. 2b) and site 2 (fig. 2c and fig. 2g) seems to be ok, given the chemical

environment. Both appear to be in or close to the selectivity filter of KdpA, which is known to bind potassium, and both spheres are coordinating to residues that might stabilize a positive charge.

However site 3 (fig. 2d) appears extremely unlikely as it is presented now. Apparently this positive ion is coordinated by the two apolar residues Ala227 and Leu228 and nothing else? I would like to see a much clearer picture of this site, and a discussion of the chemical environment of this last ion. It seems impossible to me that an ion would be fixed in this position in about ~220k particles (of ~535k total) without any kind of chemical interaction to help keep it there?

Hopefully the new supplemental figure helps to answer the raised questions. Additionally we want to point out that the position of the third ion is the weakest one and we also state this in the text now.

Furthermore the energy cost of burying a positive charge here in the middle of the membrane without clear coordination needs to be discussed. This would seem to me to have a very high energy cost, given our knowledge from potassium channels where previous work has shown that the K^+ needs to be hydrated or in a very specific selectivity filter.

It would be interesting to calculate the cost of this burial, using e.g. MD. As the radius of a hydrated K^+ ion does not fit with the tunnel, it must be at least partially dehydrated in the suggested position.

Partial hydration of the K^+ ions in the outward-facing tunnel is very probable, since the tunnel is much wider than the radius of a single potassium ion. Indeed MD calculations on K^+ ions within the tunnel are important future experiments, but beyond the scope of this work.

The cost of burying a single positively charged ion in the membrane is theoretically in the order of 40 kcal/mol (Honig et al, ann rev biophys biophys chem 15 (1986))(for reference, ATP hydrolysis releases ~12 kcal/mol in a normal cell).

Since the tunnel can create some partial hydration, the cost will be less, but still significant. The authors suggest that the tunnel and the filter are at all times loaded with potassium and ready for the next round of transport so to speak, meaning that the tunnel would presumably always have at least one buried ion within it. The energetic strain on the system for this to be correct seems to make this part of the suggested model highly unlikely. The authors should discuss this aspect of their model.

We apologize for having some difficulty in following the reviewer's argumentation. A quick check on the cited reference reveals that the reported 40 kcal/mol is the theoretical cost if a cation would cross a bilayer on its own. As a consequence, phospholipid membranes are technically impermeable to ions. Therefore, membrane proteins are required to lower the energy barrier by creating a polar environment within the membrane, through which ions can diffuse or be actively transported. This is however the case for any membrane protein that facilitates the passage of a substrate across the membrane. In particular for channels, which are not fueled by a direct source of energy as ATP hydrolysis. The determination of the energetic restraint and thus the exact amount of hydrated potassium ions that can be simultaneously located along the selectivity filter and the tunnel would be the basis of a great computational study supported by extensive MD simulations. This is, however, beyond the scope of this work and should be best conducted on the high-resolution X-ray structure, where side-chains are modeled with higher confidence as required for an exact calculation.

I am sure the authors already know this, but for the sake of the argument, I would like to point out that the refinement done in phenix.real_space_refinement does not include an electrostatic term to its target function, and therefore the refinement does not take into account the specific charge of any residues or ions. Only geometrical parameters (bond lengths, angles etc) are optimized.

Thus the model refinement does not test if the chemical environment of the site 3 (fig. 2d) K^+ ion is a stable one. Refinement of the model using MDFF, with explicit charges of the potassium comes to mind, as this might say something about the stability of an ion in this

unusual position. Have the authors considered trying this?

We agree with reviewer 5 that phenix.real_space_refinement does not take into account the electrostatic environment of the potential potassium ions and thus remains an uncertainty about the coordination of the ions. To further address this question will be part of extensive MD simulations, which initially will also include MDFF. Furthermore, anomalous signals from x-ray crystallography are desirable to better resolve the coordination sites. We would like to mention again that for us the additional densities within the entrance tunnel are an interesting observation supporting our hypothesis on the translocation pathway, but in principle we believe that the observation of inward- and outward-facing half-tunnels is already sufficient to raise this hypothesis. We now better separated the general description of the tunnels in both states (lines 180ff.) from the additional densities clearly mentioning the remaining uncertainties with respect to the potassium ion assignment (lines 214ff.).

Another related thing that should be addressed is the observation that in the EM models the 'canonical' KdpB site next to M4 is apparently empty. In the crystal structure a peak was observed here, modeled as a water. Based on those model and analysis this peak must likely be potassium instead. So, why do the authors think there is no K^+ in this position if state 1 follows the crystal conformational state (as per page 8 discussion). This discrepancy should be discussed. More generally the authors need to address why they apparently observe a K^+ ion in a very chemically unfavorable site and no K^+ ion in the adjacent much more favorable 'canonical' site of KdpB? To help the argumentation, a figure with density of this particular area would be useful. This could be part of fig. 2.

Concerning the crystal structure: To our knowledge anomalous signals were recorded at around a wavelength of 1 Å, which is not optimal for detecting K^+ anomalous signals. We would be very curious to see the anomalous signals recorded at 2 Å to further investigate the absence or presence of K^+ in the canonical binding site. Our rationale for placing state 1 after the crystal structure in the reaction cycle is that the latter is described as inhibited E1 state with a single K^+ ion in its selectivity filter, while state 1 most likely has a nucleotide-analogue bound and several K^+ ions within the outward-facing inter-subunit tunnel. We think that the tunnel we observe in state 1 is not fully opened yet (see restriction right before canonical site in Supplementary Figure 10a for state 1) and thus only partially loaded, which is why we do not see a K^+ ion at the canonical ion binding site in KdpB. We have included Figure 4c that shows the canonical binding sites of state 1 and state 2, respectively.

In state 2 the authors observe a collapse for the tunnel directly related to the breakage of the coupling helix to the P domain, and the very static KdpA structure (fig. 5a). This is the major finding of state 2, and very interesting. Furthermore, the exit tunnel/pathway observed in state 2 is not the classical entry/exit pathway known from other P-type ATPases, but a completely new path from the M4 site to the cytosol. This interesting aspect of the model should be addressed more clearly and the figure improved. It is currently very hard to follow clearly the exit tunnel in the figures. It is currently unclear how the ATPase architecture that couples ATP hydrolysis to gate opening/closing could have changed so dramatically from other P-type ATPases, and this should be discussed.

Figure 4 now has improved panels tracing the exit site more clearly. Additionally, we here detailed the structural changes in the canonical binding site. The reviewer is right in stating that the exit tunnel is not comparable to the Ca^{2+} entry or H^+ exit tunnel described as N and C path (Bublitz et al., 2013, JBC) in SERCA. But the reason for this could simply be that KdpFABC lacks TM 8-10 of type II P-type ATPases and instead its exit tunnel is formed by parts of both KdpB and KdpA. This is now mentioned in the manuscript (lines 210ff.).

Finally, I would like to argue that the manuscript does not do full justice to the copious previous literature in the field. The biochemical literature in the field need to be referenced more, since the proposed model does not really agree with most earlier biochemical studies as far as I can tell.

We included the mentioned papers and several more with a particular focus on different mutations screened and were surprised to see how good the mutations actually fit our proposed model as detailed below and in the revised manuscript.

Some of the literature that the manuscript would benefit from discussing include:

- Bramkamp et al. Biochem 44 (2005): Mutation of residues 583 and 586 of KdpB creates uncoupling of ATPase activity and transport. This indicates that these residues that are in the tunnel are part of the coupling from binding site to ATP hydrolysis site, but not part of the translocation pathway. How does this fit the presented model?

Based on the reviewer's suggestion we actually had a closer look at the canonical binding site, in which proximity both residues are located. We found that both residues significantly change their orientation comparing both states with each other: In state 1 D583 lines the entrance tunnel and may actually attract potassium ions to the canonical binding site, while in E2 state it closes the entrance tunnel. In contrast, K586 points away from the binding site in state 1 while it protrudes inside the binding site in state 2 such that it literally might push the potassium ion off the binding site (cf. Fig. 4). We hypothesize in the revised manuscript that mutations on D583 to a neutral or positively charged amino acid might actually mimic the potassium ion binding, which stimulates ATP hydrolysis but of course hinders potassium ion transport (lines 206ff.).

- Becker et al Biochem 46 (2007): Same point as above examined in more detail.

Bertrand et al J bacteriol 186 (2004): Mutations in KdpA changes the cation affinity of the complex. Does this mean that the proposed 'canonical' site of KdpB does not distinguish between ions? All other P-types have extremely strong selectivity in the canonical site. The authors should discuss why they believe the KdpB does not need such a feature.

To our knowledge not all P-type ATPases have an extremely strong selectivity in the canonical binding site as reviewed in "P-type ATPases at a glance" by Bublitz et al. Instead, actually a selectivity filter-like selection has been postulated in some cases. We believe that particularly the observation that mutations within the selectivity filter only provided Rb^{+} - or NH_4^{+} -coupled ATPase activity but not for example a Na^{+} dependency, which would have been expected in comparison to SKT members KtrB and TrkH, support our model. We speculate that upon mutating the selectivity filter, Rb^{+} and NH_4^{+} can enter the tunnel and bind to the canonical binding site because of a similar ion radius, as it has been shown for other P-type ATPases, while Na^{+} is likely not forwarded through the tunnel or bound to the canonical binding site due to poor coordination. This argumentation is detailed in the revised manuscript (lines 244ff.).

- Buurman JBC 270 (1995): They screened all Kdp subunits and identified mutations that affect substrate specificity. These were all clustered in KdpA.

See above comments.

- Dorus et al JBC 276 (2001): They saw the same thing, but focused on KdpA. Similar point as above. For these 3 papers above on selectivity, try to explain the following: In an active transporter substrate selection must be happening in the binding site that becomes occluded to ensure tight coupling. A KdpA 'prefilter' that allows both K^{+} and water to pass would not work, because then water would be enough to trigger ATP hydrolysis when bound to the KdpB binding site.

For any highly coupled active transport (like Kdp potassium transport), only when the correct substrate is bound can occlusion and transport take place. This is a fundamental principle of coupling in P-type ATPases and generally for active transport (reviewed numerous times). Do the authors think that Kdp has a different mechanism to achieve coupling, and if so, what could it be?

See above comments.

- Puppe et al. Mol Microbiol 6 (1992): They showed the KdpB Asp300 mutant increased ATPase activity and decreased K^{+} ion affinity, signs of uncoupling. This residue is part of the link to the D3 'coupling' helix in the Huang et al paper. Explain this observation using the proposed model. In this model the coupling helix is static and breaking this bond thus might free the ATPase to hydrolyze ATP faster, but why would K^{+} affinity decrease and uncoupling occur?

To be precise the mutations tested in the paper all showed an increased K_m for K^+ -stimulated ATPase activity while the growth phenotype was not affected actually suggesting a not changed K_m for K^+ transport. Thus the authors of the paper speculated that D300 might be a regulatory binding site for potassium. We think we cannot exclude this hypothesis by now and the fact that D300 is located within the here suggested pathway might actually strengthen such regulatory function. However, further research is required to elucidate the coupling mechanism of potassium ion binding to the canonical binding site and K^+ release to ATP hydrolysis, autophosphorylation and autodephosphorylation. Nonetheless, we at least mention the mutation in the revised version of our manuscript (line 256ff.).

In conclusion, I have a lot of questions and some concerns, especially on the proposed model, but in the end, this is for the authors and others to challenge with future research, and will set the stage for a lot of interesting discussions in the field. Overall the methodology is sound and the findings highly interesting, and I would again recommend publication after some edits and clarifications. I am looking forward to discussing these fascinating results with the authors at future conferences. Congratulations again! Thanks again for the kind words and particularly for the great input!

Reviewers' Comments:

Reviewer #1:

Remarks to the Author:

The revised version of the manuscript represents an appreciated improvement compared to the initial version. The presentation of data and the discussion of the results provide now a clear insight into the approach and the line of arguments that led to the proposal of the new and very interesting K⁺-transport mechanism.

While the authors' response to the second issue of this reviewer is satisfying, I would like to stress again the first one to get my intentions across more clearly in favor of an indicative discussion concerning a further development of the transport mechanism.

The classification of specifiable states of the KdpFABC complex is indeed curtailed by the rather low resolution of 4-6 Å in the cytoplasmic domains and of 3.7 Å at its best in the TMD. Therefore, the subdivision into two conformational states, based on the feasible discrimination of the spatial arrangement of N, P, and A domain is justified and well supported by the distance distribution obtained from EPR measurements. In the desire to advance our understanding of a structure-function relation of the K⁺-transporting complex, one should, however, keep in mind issues which were presented by the authors (and in the literature) that blur somehow the conclusions drawn from the shown structures at the present state of investigations: (1) According to Fig. 5, the KdpFABC complex can adopt one of five defined, consecutive states of the Post-Albers cycle under the chosen experimental conditions, although by computational constraints all analyzed images ended up in what was introduced as the first or last state of that sequence. (2) Since it is known that there are two principal conformations in P-type ATPases, E₁ and E₂, and taking into account the low experimental resolution and the "flexibility" of the complex, it is not surprising that most of the images were merged in only two states. (3) But this does in no way exclude that different particular protein "sub-states" of E₁ and E₂ are embraced in either of the presented "states", because the images were sorted preferentially by the arrangement of the N, P, and A domains and not by additional "too small" components such as ions or nucleotides. (4) Nevertheless, the faint densities of in places where those components may be expected should be taken and acknowledged as serious clues for a greater diversity of structures. My concern is not that the presented structural information was not reliable. But you rather should point out that you are aware of the fact that you can present only part of story at the moment although you may hint that there is more. In the worst case one or both of your presented structures are inhibited states that are not an essential part of the actual states composing the physiological pump cycle. It would help the readers and further discussions of the matter to keep an open mind for this aspect. I recommend emphatically adding a few appropriate statements.

Reviewer #2:

Remarks to the Author:

I am fully satisfied with the changes made to the manuscript and recommend its acceptance.

Reviewer #3:

Remarks to the Author:

While the revised manuscript by Stock et al. is improved compared to the original version in terms of presentation and discussion of previously known mechanistic insights from the literature, the authors have decided not to add any biophysical or functional data to strengthen their mechanistic conclusions. While the structural analysis remains solid and interesting, we believe the lack of direct functional insight in the form of mutagenic/transport assays remains problematic because there is no convincing evidence regarding that the channel component remains closed during the transport cycle. Our suggestions regarding functional data have been deemed outside the scope of

the current manuscript or rebutted by additional speculation and reliance on an argument of overall ambiguity/need for further experiments existing in the field (as it relates to the K⁺ traversal pathway through the complex). While some of the overly speculative statements in the original manuscript have been removed / toned down, the authors still over-interpret / sell their "new" mechanism without supportive evidence. In consequence, the mechanistic interpretations are far from conclusive. The problem is inherent in the paper and noticeable in many places. For example, the word "reveal" in the title is evidently misleading.

A recommendation for publication of this study in its current form can only be based on the structures presented and in recognition of the limited merits of the functional and mechanistic conclusions drawn.

Reviewer #4:

Remarks to the Author:

The authors have addressed the concerns raised in the review and the manuscript is now suitable for publication.

Reviewer #5:

Remarks to the Author:

The authors have provided a heavily modified manuscript with new text and figures.

I am happy to say that most of my concerns have been addressed in a careful manner, and the new text is much easier to read and follow. The figures have been much improved over the original which again helps in showcasing the results.

I only have a few comments and suggestions, and otherwise find the manuscript ready for publication.

Comments.

1) The use of the word chimera in the abstract and elsewhere is unorthodox, and like reviewer #2, I suggest the authors change it.

2) The introduction is generally good. One sentence (line 58) states "unites a P-type ATPase with an ion channel".

KdpA is not an ion channel. It is an channel-like SKT protein family member. The statement will confuse the uninitiated reader, and upset physiologists in particular I predict (based on personal experience).

3) line 84. "To enlighten the mechanism" is not proper English. The sentence should be rephrased.

4) line 88 and elsewhere: "SERCA" is never defined nor explained. This should be done at least the first time. eg. The Calcium P-type ATPase SERCA.

5) line 111ff: I appreciate the new discussion of the phosphorylation of S162 and whether it inhibits ATPase activity and turnover. In the Huang et al Nature (2017) paper, the phosphorylation itself was suggested to stop turnover, and shown to inhibit ATPase activity. The interaction with residues Lys357 and Arg363 of the N-domain was not a strict requirement for this suggestion and observation. This is also highlighted by the fact that neither of the two positive residues are conserved in sequence in other Kdp sequences. Thus saying that the new results contradict the previously proposed model reads as a bit of an overstatement. In fact they confirm the finding that S162 is phosphorylated in purified KdpFABC, which is a very puzzling feature.

6) The EPS experiments are much better described. I still do not quite follow the authors argument (line 133ff) that theoretical state 1 resembles the blue trace (fig 2c) better than the top cyan trace (btw the color blue and cyan are too similar and I suggest to change one of them). Arguable the theoretical crystal trace in the top panel resembles the blue trace more than the cyan trace (even more so than the state 1 trace), if an X axis shift can be dismissed. Nevertheless that data is much easier to read and evaluate now, and this train of thought can be left as an exercise to the reader.

7) I much appreciate the new formulation of text explaining the docking of inhibitors into the structures. This is a key improvement.

8) Likewise the explanation of how the domains are aligned compared to SERCA is significantly improved, and the separation of tunnels from the proposed K⁺ densities is very helpful for the reader.

9) The RMSD calculations are not well defined. How many atoms were used in each case (C-alpha, or all atoms etc, is it done only using the cytosolic domains of KdpB or the full model)? To provide a clear comparison, the A, N and P domains could for instance be aligned individually first to SERCA, to provide a "best possible fit" score, since the RMSD could never be 0 Å if the models are different. Overall the scores are all quite bad. ~3 Å is normally not considered a good fit between models. For this reason the 'best possible fit' would be nice to know. Clearly the E2 SERCA models fit best with state 2 while the E1 SERCA models fit best with state 1, so the results help the argumentation of the conformation of the observed states.

10) While the figures are much improved, I do miss a main figure that shows the map, as this is the result, and the model is the authors interpretation of their result. I suggest this is made as an additional panel of figure 1. This panel could be quite similar to fig1a of the original draft (but the states shown side by side) or Supp fig 2 c,d. Another example of how this can be done can be seen in Gong et al Nature Comm 9:3623 (2018) figure 1a. If one only saw the model from fig1c in that paper, it would be hard to evaluate the quality of the NPTN model.

12) Line 217: "water ions are not visible at 3.7 Å". Water is not an ion first of all. Secondly an oxygen atom is visible at 3.7 Å if it is well ordered (eg. carbonyl group of the protein main chain). Water is not generally visible because it is not ordered enough to be observed at this resolution. Blanket statements like this one should be avoided in the text.

13) Thank you for supp fig 11. This gave a clearer picture of the data on the ion-peaks.

14) The putative lipid density found close to peak 3 (line 229). It would be useful to have a measure of the distance from the lipid to the ion (in the figure). A better description of how the authors envision the hydrophobic tail of a lipid coordinate the charged K⁺-ion as written in line 230 would also be relevant.

How sure are the authors that the peak is not part of the lipid? We often see lipid densities with peaks at the end of the aliphatic tails. I have attached a screendump from an ongoing EM-structure we are working on, with a modeled lipid and the 3Å map contoured at 9 sigma, to showcase this.

15) The analysis the authors did on the identity of the peaks (by blurring maps) is fascinating and should be part of the manuscript as a supplementary figure. It is a very interesting finding, and it will be very helpful for other people in the field to know that exceptions to the Wang et al. IUCrJ 5 (2018) paper conclusions exist. The argument that the selectivity filter is the cause of the difference is hard to follow since the site 3 peak (fig 3d) behaves the same way. The site 1 and 2 peaks (fig 3 b,c) seem reasonable as K⁺ while peak 3 is somewhat more doubtful. The authors say they state this in the text, but I could not find the statement?

"(page 21) we want to point out that the position of the third ion is the weakest one and we also

state
this in the text now.”

16) Affinity. The authors need to be clear about affinity vs. selectivity in the manuscript. More specifically, the authors need to explain clearly where they believe μM K-affinity is created in the system. Sometimes they say it is the selectivity filter of KdpA (line 240, 272, 287) and sometimes the canonical site of KdpB seems to be suggested (line 45 (indirectly), 273). In this context they should speculate on why the KdpA Q116R mutant lower the K_m value of the system from μM to mM . Since KdpA is static, this must mean that the μM affinity is derived from the selectivity filter site? Affinity in the canonical KdpB site must arguably then be in the mM range (due to the Q166 mutant biochemistry). Since no change happens to the selectivity filter during the proposed transport cycle, it would be interesting to discuss how the ion move from a μM KdpA site to a mM KdpB site, if the authors believe this to be the case. As I understand the text currently, this is the proposed model?

The dynamic and low affinity pre-filters proposed in other P-type ATPase systems have 1) never been shown to exist, but has been speculated upon in Bublitz (2011) and Morth (2011), and with some experimental hints (Einholm JBC 282 (2007), Einholm JBB 39 (2007), Laursen JBC 284 (2009)) and, more importantly, 2) would be dynamic and change with the conformational movements of the pump, and 3) have always been suggested to confer selectivity, not affinity. The pump prefilter hypothesis is not similar to the suggested mechanism for Kdp, which would be a fundamental new look on P-type ATPase mechanism, and part of the attraction and potential impact of this manuscript.

17) Line 246: HKT is not defined.

Reviewers' comments/authors' replay:

Reviewer #1 (Remarks to the Author):

The revised version of the manuscript represents an appreciated improvement compared to the initial version. The presentation of data and the discussion of the results provide now a clear insight into the approach and the line of arguments that led to the proposal of the new and very interesting K⁺-transport mechanism.

While the authors' response to the second issue of this reviewer is satisfying, I would like to stress again the first one to get my intensions across more clearly in favor of an indicative discussion concerning a further development of the transport mechanism.

The classification of specifiable states of the KdpFABC complex is indeed curtailed by the rather low resolution of 4-6 Å in the cytoplasmic domains and of 3.7 Å at its best in the TMD. Therefore, the subdivision into two conformational states, based on the feasible discrimination of the spatial arrangement of N, P, and A domain is justified and well supported by the distance distribution obtained from EPR measurements. In the desire to advance our understanding of a structure-function relation of the K⁺-transporting complex, one should, however, keep in mind issues which were presented by the authors (and in the literature) that blur somehow the conclusions drawn from the shown structures at the present state of investigations: (1) According to Fig. 5, the KdpFABC complex can adopt one of five defined, consecutive states of the Post-Albers cycle under the chosen experimental conditions, although by computational constraints all analyzed images ended up in what was introduced as the first or last state of that sequence. (2) Since it is known that there are two principal conformations in P-type ATPases, E₁ and E₂, and taking into account the low experimental resolution and the "flexibility" of the complex, it is not surprising that most of the images were merged in only two states. (3) But this does in no way exclude that different particular protein "sub-states" of E₁ and E₂ are embraced in either of the presented "states", because the images were sorted preferentially by the arrangement of the N, P, and A domains and not by additional "too small" components such as ions or nucleotides. (4) Nevertheless, the faint densities of in places where those components may be expected should be taken and acknowledged as serious clues for a greater diversity of structures. My concern is not that the presented structural information was not reliable. But you rather should point out that you are aware of the fact that you can present only part of story at the moment although you may hint that there is more. In the worst case one or both of your presented structures are inhibited states that are not an essential part of the actual states composing the physiological pump cycle. It would help the readers and further discussions of the matter to keep an open mind for this aspect. I recommend emphatically adding a few appropriate statements.

Thanks for the suggestion, to which we agree. To increase awareness that the final reconstruction might contain a subset of particles in a slightly different conformation of the N, P and A domain we have rephrased the paragraphs starting from line 135 and line 179.

In fact, there are certainly more than the 4 sub-states we have sketched in Figure 5, and as stated in lines 387ff. we do not believe that any of the three available structures represent exactly those cartoons. We are looking forward to (us or others) being able to solve in more detail additional E1 and E2 intermediates, as this will be perquisite to truly understand the mechanism of KdpFABC.

Reviewer #2 (Remarks to the Author):

I am fully satisfied with the changes made to the manuscript and recommend its acceptance.

Reviewer #3 (Remarks to the Author):

While the revised manuscript by Stock et al. is improved compared to the original version in terms of presentation and discussion of previously known mechanistic insights from the literature, the authors have decided not to add any biophysical or functional data to strengthen their mechanistic conclusions. While the structural analysis remains solid and interesting, we believe the lack of direct functional

insight in the form of mutagenic/transport assays remains problematic because there is no convincing evidence regarding that the channel component remains closed during the transport cycle. Our suggestions regarding functional data have been deemed outside the scope of the current manuscript or rebutted by additional speculation and reliance on an argument of overall ambiguity/need for further experiments existing in the field (as it relates to the K⁺ traversal pathway through the complex). While some of the overly speculative statements in the original manuscript have been removed / toned down, the authors still over-interpret / sell their “new” mechanism without supportive evidence. In consequence, the mechanistic interpretations are far from conclusive. The problem is inherent in the paper and noticeable in many places. For example, the word “reveal” in the title is evidently misleading. A recommendation for publication of this study in its current form can only be based on the structures presented and in recognition of the limited merits of the functional and mechanistic conclusions drawn.

Thanks for appreciating the improvement of the manuscript. To provide an additional biochemical experiment, which indicates that KdpA does not open in a channel-like manner and cannot translocate K⁺ by itself, we have now included a complementation assay. As discussed in lines 284 ff., in the new supplementary figure 12 we now show growth complementation assays of *E. coli* LB2003, a strain deficient in all endogenous K⁺ uptake systems, complemented with the entire KdpFABC complex, with only the KdpA subunit and finally with the channel-like subunit KtrB of the analogous KtrAB complex, respectively. While the expression of KdpFABC and the channel-like KtrB subunit are sufficient to complement the strain and allow cell growth at low potassium concentrations, KdpA-alone expressing cells do not grow below 10 mM K⁺. The results confirm, that while a structural comparison would suggest that KdpA might act like a channel, it remains impermeable in the absence of the other subunits in contrast to its paralog KtrB. We acknowledge that this is still not a solid biochemical prove that KdpA might never open in a channel-like manner within the complex. However, we are certain that the reviewer agrees that it is harder to prove something that never happens than something that happens. Along these lines we would equally like to stress out that to our knowledge, among all tested mutations and biochemically studies there was never a prove that KdpA opens in a channel-like manner either. In contrast, all available mutations which impaired transport are located in the selectivity filter (upper part of KdpA) or near the canonical binding site in KdpB, supporting our proposed mechanism. Further, as pointed out by the reviewer, the here provided structural data in comparison with the available X-ray structure provide a very strong argument towards our conclusions. We aim to design further mutagenesis studies to confirm this hypothesis and are happy to include them in a future manuscript. In order to clarify that the proposed mechanism is just a structure-based model we have made various changes in the manuscript (see for example the title, the abstract and the conclusion).

Reviewer #4 (Remarks to the Author):

The authors have addressed the concerns raised in the review and the manuscript is now suitable for publication.

Reviewer #5 (Remarks to the Author):

The authors have provided a heavily modified manuscript with new text and figures.

I am happy to say that most of my concerns have been addressed in a careful manner, and the new text is much easier to read and follow. The figures have been much improved over the original which again helps in showcasing the results.

I only have a few comments and suggestions, and otherwise find the manuscript ready for publication.

Comments.

1) The use of the word chimera in the abstract and elsewhere is unorthodox, and like reviewer #2, I suggest the authors change it.

Changed to 'chimeric complex' which is more commonly used.

2) The introduction is generally good. One sentence (line 58) states "unites a P-type ATPase with an ion channel".

KdpA is not an ion channel. It is an channel-like SKT protein family member. The statement will confuse the uninitiated reader, and upset physiologists in particular I predict (based on personal experience).

Changed to 'channel-like protein'

3) line 84. "To enlighten the mechanism" is not proper English. The sentence should be rephrased.

Changed to 'To elucidate'

4) line 88 and elsewhere: "SERCA" is never defined nor explained. This should be done at least the first time. eg. The Calcium P-type ATPase SERCA.

Changed as suggested

5) line 111ff: I appreciate the new discussion of the phosphorylation of S162 and whether it inhibits ATPase activity and turnover. In the Huang et al Nature (2017) paper, the phosphorylation itself was suggested to stop turnover, and shown to inhibit ATPase activity. The interaction with residues Lys357 and Arg363 of the N-domain was not a strict requirement for this suggestion and observation. This is also highlighted by the fact that neither of the two positive residues are conserved in sequence in other Kdp sequences. Thus saying that the new results contradict the previously proposed model reads as a bit of an overstatement. In fact they confirm the finding that S162 is phosphorylated in purified KdpFABC, which is a very puzzling feature.

Modified as follows:

"Thus, the phosphorylation of Ser162 in KdpB does not lock the A and N domains in an auto-inhibited conformation, as previously suggested⁴, and it remains elusive whether the phosphorylation fully inactivates KdpFABC or significantly slows down its activity."

6) The EPS experiments are much better described. I still do not quite follow the authors argument (line 133ff) that theoretical state 1 resembles the blue trace (fig 2c) better than the top cyan trace (btw the color blue and cyan are too similar and I suggest to change one of them). Arguable the theoretical crystal trace in the top panel resembles the blue trace more than the cyan trace (even more so than the state 1 trace), if an X axis shift can be dismissed. Nevertheless that data is much easier to read and evaluate now, and this train of thought can be left as an exercise to the reader.

We fully agree with the reviewer that we cannot be 100% sure about this point, but in the last revised version of the manuscript we already acknowledge and discuss this discrepancy in detail, allowing the reader to judge the data. However, since we obtain two main states in cryo-EM and in the EPR measurements upon the addition of the inhibitor mix, we feel confident that our assumption is correct. Future higher resolution structures will answer this question.

7) I much appreciate the new formulation of text explaining the docking of inhibitors into the structures. This is a key improvement.

8) Likewise the explanation of how the domains are aligned compared to SERCA is significantly improved, and the separation of tunnels from the proposed K⁺ densities is very helpful for the reader.

9) The RMSD calculations are not well defined. How many atoms were used in each case (C-alpha, or all atoms etc, is it done only using the cytosolic domains of KdpB or the full model)?

To provide a clear comparison, the A, N and P domains could for instance be aligned individually first to SERCA, to provide a "best possible fit" score, since the RMSD could never be 0 Å the models are

different. Overall the scores are all quite bad. $\sim 3 \text{ \AA}$ is normally not considered a good fit between models. For this reason the 'best possible fit' would be nice to know. Clearly the E2 SERCA models fit best with state 2 while the E1 SERCA models fit best with state 1, so the results help the argumentation of the conformation of the observed states.

A more detailed explanation has been added and RMSD values for the best possible fit are provided. In addition we repeated the RMSD calculation just based on the NPA domains instead of using the whole KdpB subunit, as we think these data are more reliable. This way most RMSD values slightly dropped. However, we believe that an RMSD value in the 2–4 Å range is what we would expect for a comparison between similar but yet distantly related proteins.

10) While the figures are much improved, I do miss a main figure that shows the map, as this is the result, and the model is the authors interpretation of their result. I suggest this is made as an additional panel of figure 1. This panel could be quite similar to fig1a of the original draft (but the states shown side by side) or Supp fig 2 c,d. Another example of how this can be done can be seen in Gong et al Nature Comm 9:3623 (2018) figure 1a. If one only saw the model from fig1c in that paper, it would be hard to evaluate the quality of the NPTN model.

We have included maps of both states in revised figure 1.

12) Line 217: “water ions are not visible at 3.7 \AA ”. Water is not an ion first of all. Secondly an oxygen atom is visible at 3.7 \AA if it is well ordered (eg. carbonyl group of the protein main chain). Water is not generally visible because it is not ordered enough to be observed at this resolution. Blanket statements like this one should be avoided in the text.

We truly thank the reviewer for this unintentional mistake; obviously we are aware that water is not an ion. The whole statement has been removed. Instead, we based our hypothesis of potassium ions being bound in the tunnel on the high selectivity for K^+ and the addition of 1 mM KCl to the sample (lines 259f).

13) Thank you for supp fig 11. This gave a clearer picture of the data on the ion-peaks.

14) The putative lipid density found close to peak 3 (line 229). It would be useful to have a measure of the distance from the lipid to the ion (in the figure). A better description of how the authors envision the hydrophobic tail of a lipid coordinate the charged K^+ -ion as written in line 230 would also be relevant.

How sure are the authors that the peak is not part of the lipid? We often see lipid densities with peaks at the end of the aliphatic tails. I have attached a screendump from an ongoing EM-structure we are working on, with a modeled lipid and the 3 \AA map contoured at 9 sigma, to showcase this.

We realized that this comment was a bit far fetched. We changed the according sentence to “In addition, we found an unassigned density at the interface of KdpA and KdpB that most likely corresponds to a bound lipid, which might contribute to tunnel formation and ion propagation (Supplementary Figure 11e and f).” (lines 276ff.). At this stage it is impossible to predict how the putative lipid might affect ion binding or translocation (probably it does not directly bind the ion), which is why we do not want to over-interpret the data. We prefer to simply point out the existence of the clear density and its prominent location, which might suggest a functional role.

We do not expect the peak we assigned as potassium ion to be part of the lipid mostly because it is missing in the E2 state. Furthermore, the densities are clearly separated.

15) The analysis the authors did on the identity of the peaks (by blurring maps) is fascinating and should be part of the manuscript as a supplementary figure. It is a very interesting finding, and it will be very helpful for other people in the field to know that exceptions to the Wang et al. IUCrJ 5 (2018) paper conclusions exist.

Since we do not believe this discussion helps to follow the argumentation of the paper, we would rather not include this discussion in the paper. However, we will agree on publishing the reviewing process such that it will be available to the public, which will also provide the right context.

The argument that the selectivity filter is the cause of the difference is hard to follow since the site 3 peak (fig 3d) behaves the same way. The site 1 and 2 peaks (fig 3 b,c) seem reasonable as K⁺ while peak 3 is somewhat more doubtful. The authors say they state this in the text, but I could not find the statement?

“(page 21) we want to point out that the position of the third ion is the weakest one and we also state this in the text now.”

We have it now included in lines 274ff. “...while only the hydroxyl group of Ala227 seems to be in direct contact with the third ion, which shows the weakest density.”

16) Affinity. The authors need to be clear about affinity vs. selectivity in the manuscript. More specifically, the authors need to explain clearly where they believe μM K-affinity is created in the system. Sometimes they say it is the selectivity filter of KdpA (line 240, 272, 287) and sometimes the canonical site of KdpB seems to be suggested (line 45 (indirectly), 273). In this context they should speculate on why the KdpA Q116R mutant lower the K_m value of the system from μM to mM . Since KdpA is static, this must mean that the μM affinity is derived from the selectivity filter site? Affinity in the canonical KdpB site must arguably then be in the mM range (due to the Q166 mutant biochemistry). Since no change happens to the selectivity filter during the proposed transport cycle, it would be interesting to discuss how the ion move from a μM KdpA site to a mM KdpB site, if the authors believe this to be the case. As I understand the text currently, this is the proposed model? The dynamic and low affinity pre-filters proposed in other P-type ATPase systems have 1) never been shown to exist, but has been speculated upon in Bublitz (2011) and Morth (2011), and with some experimental hints (Einholm JBC 282 (2007), Einholm JBB 39 (2007), Laursen JBC 284 (2009)) and, more importantly, 2) would be dynamic and change with the conformational movements of the pump, and 3) have always been suggested to confer selectivity, not affinity. The pump prefilter hypothesis is not similar to the suggested mechanism for Kdp, which would be a fundamental new look on P-type ATPase mechanism, and part of the attraction and potential impact of this manuscript.

Indeed, in particular the Q116R mutant raises questions on how, not the selectivity but the high affinity is built in the complex. We acknowledge that a 100fold increase in K_m in Q116R might suggest that the affinity in the canonical binding site is in a millimolar range. However, the introduction of an arginine at the entrance of the selectivity filter could equally represent a sterically and electrostatic barrier before the main selectivity filter. Further, in an assay where not single steps but rather the final potassium uptake is measured the calculated K_m value represents the lowest binding affinity or an even slower, reaction step [$K_M = ((\frac{k_1}{k_2} + \frac{k_1}{k_3}) / \frac{k_1}{k_4})$ and $K_D = (\frac{k_1}{k_2} / \frac{k_1}{k_4})$, where k_2 describes a second reaction step]. If

we then assume a sequential process as described here, we believe that if the first step at the entrance to the selectivity filter is the rate-limiting step (here due to low affinity), one would not be able to detect a higher affinity binding site at a subsequent step. Nonetheless, we would like to emphasize that this part contains a lot of speculations. Our main finding is the identification of two half-channels, which we believe allow an actual active transport of K⁺. Additional research will be necessary to comprehensively elucidate how potassium ions can propagate through the tunnels and which roles the different binding sites play. To acknowledge this uncertainty we have added the following sentence to the manuscript (lines 310ff): “On the other hand, to which extend the selectivity filter, the tunnel and the canonical binding contribute to the observed high affinity of KdpFABC will require additional studies.”

17) Line 246: HKT is not defined.

A definition has been added.

Reviewers' Comments:

Reviewer #1:

Remarks to the Author:

The issues raised by this reviewer were considered and included satisfactorily.

Reviewer #5:

Remarks to the Author:

I am satisfied with the changes made to the manuscript and again recommend its acceptance.